# Learning CAD Modeling Sequences via Projection and Part Awareness

**Yang Liu**
Beijing Institute of Technology
China, Beijing
liuyang@bit.edu.cn

**Daxuan Ren**
Nanyang Technological University
Singapore
daxuan001@e.ntu.edu.sg

**Yijie Ding**
Nanyang Technological University
Singapore
yijie002@e.ntu.edu.sg

**Jianmin Zheng**
Nanyang Technological University
Singapore
ASJMZheng@ntu.edu.sg

**Fang Deng** *
Beijing Institute of Technology
China, Beijing
dengfang@bit.edu.cn

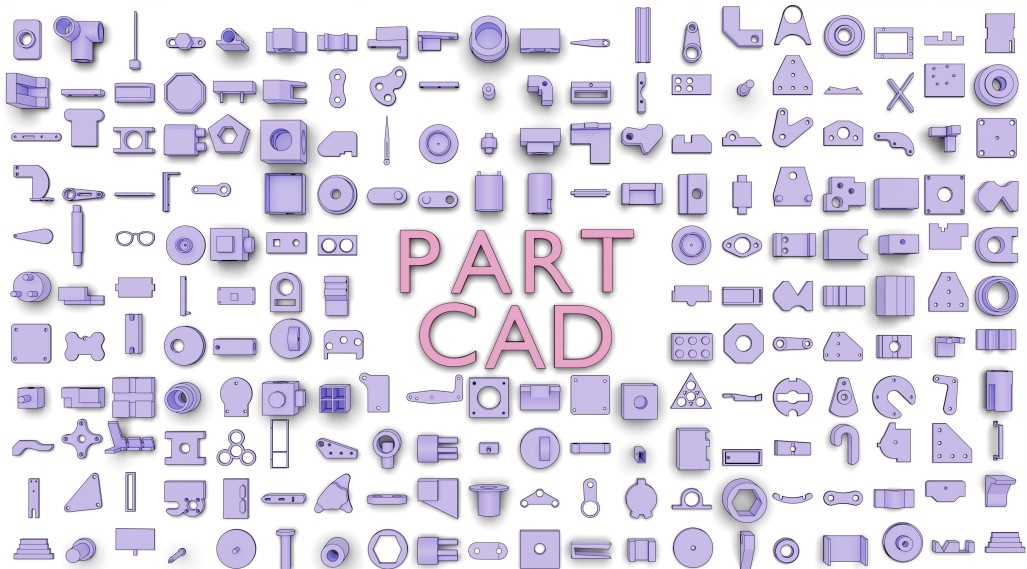

Figure 1: CAD Library reconstructed by PartCAD

## Abstract

This paper presents PartCAD, a novel framework for reconstructing CAD modeling sequences directly from point clouds by projection-guided, part-aware geometry reasoning. It consists of (1) an autoregressive approach that decomposes point clouds into part-aware latent representations, serving as interpretable anchors for

---
*Corresponding author (dengfang@bit.edu.cn)

39th Conference on Neural Information Processing Systems (NeurIPS 2025).

CAD generation; (2) a projection guidance module that provides explicit cues about underlying design intent via triplane projections; and (3) a non-autoregressive decoder to generate sketch-extrusion parameters in a single forward pass, enabling efficient and structurally coherent CAD instruction synthesis. By bridging geometric signals and semantic understanding, PartCAD tackles the challenge of reconstructing editable CAD models—capturing underlying design processes—from 3D point clouds. Extensive experiments show that PartCAD significantly outperforms existing methods for CAD instruction generation in both accuracy and robustness. The work sheds light on part-driven reconstruction of interpretable CAD models, opening new avenues in reverse engineering and CAD automation.

# 1 Introduction

Computer-Aided Design (CAD) plays a central role in modern industrial manufacturing and product development [1]. While CAD models can be represented by point clouds, meshes, boundary representations (B-Reps), and constructive solid geometry (CSG), these representations primarily capture the final geometry rather than the generative design process. In contrast, the modeling sequence representation defines the model through a sequence of modeling commands such as sketching, extrusion, and Boolean operations that encode not only geometric construction but also high-level design intent and the underlying semantic logic. These modeling sequences are fundamental for supporting editability, version control, and downstream analysis [2]. However, recovering such sequences from raw 3D data, especially from unstructured inputs like scanned point clouds, remains a challenging problem. Manual reconstruction is labor-intensive and requires substantial domain expertise [3, 4, 5]. Thus, automating the conversion from point clouds into editable modeling workflows offers transformative potential for reverse engineering, redesign, and knowledge transfer in CAD systems [6, 7].

This paper focuses on converting 3D point clouds into ordered parametric instructions—such as sketches (e.g., line, arc, circle) and operations (e.g., extrusion) [8, 9]—that can be replayed or edited in standard CAD environments [10, 11, 12]. The problem by nature lies at the intersection of geometric reasoning, program induction, and sequential learning. Recent works have framed this as a sequence prediction problem, applying encoder-decoder architectures to map geometric inputs to command sequences [13]. While effective for simple shapes, these approaches often fail to scale to real-world models that require reasoning over intermediate constructions, hierarchical dependencies, and geometric constraints. To mitigate this, other methods incorporate geometric priors or sketch-based abstractions to guide the generation process [14]. More recent hybrid models combine low-level geometric cues with high-level structure to improve interpretability and output validity, though many of them still focus on fitting continuous primitives rather than generating symbolic instructions, limiting their ability to capture procedural intent [15].

Despite substantial advances in CAD modeling sequence generation, current techniques still exhibit notable limitations. For instance, many approaches treat raw geometric input as a monolithic entity, overlooking the procedural structure and hierarchical dependencies inherent to CAD modeling, which in turn leads to inefficiencies in capturing design intent and guiding accurate reconstruction. Another challenge lies in the sequential reasoning of CAD operations. Fully autoregressive methods ensure sequential consistency but are highly vulnerable to error accumulation, where early mistakes propagate and compound over subsequent steps [16, 17]. In contrast, non-autoregressive approaches improve efficiency by generating sequences in parallel, but often struggle to capture hierarchical dependencies and structural constraints, leading to incoherent or invalid outputs [18, 19]. In addition, current methods lack explicit mechanisms to align high-level geometric features with stepwise modeling instructions, limiting their ability to produce semantically consistent instructions. The inherent one-to-many mapping between geometry and valid command sequences further introduces ambiguity, complicating both learning and inference. These challenges lead to a fundamental question: ***How can raw geometry be structured and interpreted to reflect the underlying design process and guide the generation of accurate and semantically meaningful CAD instructions?***

To answer this question, we introduce **PartCAD**, a semi-autoregressive framework that generates structured CAD modeling instructions by leveraging implicit part abstractions and projection-guided geometry. Rather than treating raw input as a whole, PartCAD autoregressively decomposes the point cloud into a sequence of part-aware latent representations, each semantically aligned with a simple modeling step, thereby providing procedural structure and capturing design intent. These

latents serve as anchors for CAD generation, guided by canonical triplane projections that explicitly encode view-aligned geometric cues. To decode latents into parametric instructions, PartCAD adopts a non-autoregressive decoding strategy that produces precise *sketch–extrusion* parameters in a single forward pass. Several key techniques are introduced to bridge low-level geometry and high-level semantics for coherent and interpretable CAD reconstruction. The main contributions of our work include:

- PartCAD, a semi-autoregressive framework for parametric CAD generation, which decomposes input point clouds into procedurally aligned part-aware latents and generates structured modeling instructions with triplane projection guidance;
- an adaptive projection strategy to refine view-aligned geometry, and a hierarchical KNN aggregation kernel to extract robust triplane features;
- a projection-guided decoding mechanism that integrates part-level semantics with both point-level local features and global geometric context, enabling accurate CAD instruction generation in a single forward pass.

Extensive experiments demonstrate that PartCAD achieves superior performance in learning CAD instruction sequences and generalizes well across domains (see Figure 1 for some models reconstructed by PartCAD). The proposed approach provides new insights into reconstructing editable and semantically meaningful CAD programs from raw geometry.

## 2  Related Works

**Shape Representations:** Various low-level shape representations such as point clouds, voxels, and meshes are widely used to describe 3D geometry. These discrete formats have become standard in modern 3D vision pipelines, supporting substantial progress in tasks like feature extraction, segmentation, and classification [20, 21, 22, 23, 24]. With advances in geometric learning, recent techniques enable translation across representations, for example, reconstructing meshes from point clouds [25, 26] or converting voxel grids into implicit fields [27, 28]. Their outputs, however, lack explicit structure and editability, and fail to convey the parametric representation needed for downstream design, analysis and manipulation. Our goal is to translate raw geometric inputs, particularly point clouds, into structured parametric representations suitable for procedural modeling.

**Parametric CAD Learning:** Parametric modeling aims to learn structured representations of CAD models, with Constructive Solid Geometry (CSG) and Boundary Representation (B-Rep) being two widely adopted paradigms. CSG constructs 3D shapes by hierarchically combining primitives (e.g., cubes, spheres, cylinders) through Boolean operations (e.g., union, intersection, difference), making it well-suited for procedural tasks involving regular geometries and well-defined topologies. Recent works have applied deep neural networks to infer CSG trees for inverse or generative design, showing promising results on synthetic or simple shapes [29, 30, 31]. However, its reliance on a limited primitive vocabulary restricts expressiveness in modeling complex surfaces and high-level design intent. In contrast, B-Rep explicitly represents geometry through vertices, edges, and faces, offering high fidelity in capturing intricate geometric and topological details. Combined with deep learning, recent methods have exploited B-Rep using graph neural networks and Transformer-based models to learn spatial and connectivity-aware features [32, 33, 34, 35, 36, 37]. Nevertheless, learning with B-Rep remains challenging due to its strict consistency constraints and sensitivity to topological variations or geometric imperfections. Moreover, B-Rep focuses on final geometry rather than the design process, limiting its capacity for sequential reasoning and intent-aware editing. Unlike these works, we focus on learning parametric instruction sequences (e.g., sketch, extrusion, Boolean operations) that describe 3D shapes through a step-by-step modeling process. These instructions compactly encode both geometric structure and design logic in an interpretable form, aligning naturally with procedural modeling workflows.

**CAD Instruction Language Generation:** Instruction language offers a procedural representation of 3D modeling, where shapes are incrementally constructed via sequences of semantic commands. Rather than directly defining geometries, CAD instruction captures the logic of the actual modeling workflows, offering high interpretability and editability [38]. Early methods used handcrafted rules, templates, or grammars to encode domain-specific priors. Recently, generative paradigms have emerged, leveraging the semantic and sequential nature of instruction sequences under supervised or

unsupervised learning [39, 40, 41]. Autoencoder-based methods encode entire instruction sequences into compact latent vectors and decode them holistically, capturing global dependencies and improving stability [42]. Seq2Seq models formulate instruction generation as conditional sequence translation from multimodal inputs (e.g., sketches, images, point clouds) to command sequences, aided by attention mechanisms for long-range dependency and cross-modal alignment [43, 44]. Autoregressive models further enhance local control and syntactic consistency by generating instructions token-by-token [45, 46, 47, 48]. Additionally, large-scale pre-trained models (e.g., large language models or multimodal large models) have been introduced into CAD instruction generation [49, 50, 51, 52, 53]. In contrast to existing paradigms, we propose a semi-autoregressive framework that first derives part-aware latent representations in an autoregressive manner, followed by a projection-guided non-autoregressive decoder to generate CAD instructions in a single forward pass. This design enables precise semantic reasoning alongside efficient and structurally coherent instruction generation.

# 3 PartCAD for Inferring Modeling Sequences from Point Clouds

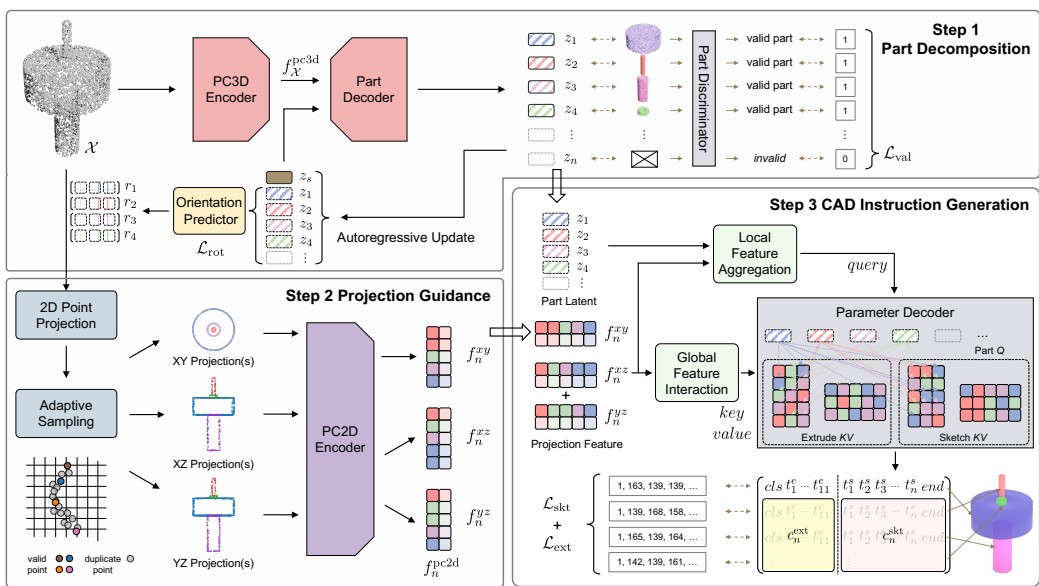

Figure 2: **Overview of PartCAD Architecture.** Starting from a point cloud $\mathcal{X}$, PartCAD generates parametric CAD instructions in three steps: (1) *Part Decomposition* extracts a global point cloud feature $f_{\mathcal{X}}^{\mathrm{pc3d}}$ and autoregressively decodes it into a sequence of part latents $z_1, \ldots, z_N$; (2) *Projection Guidance* predicts an orientation $r_n$ for each $z_n$ and derives canonical-aligned projections used to extract triplane features $f_n^{\mathrm{pc2d}} = \{f_n^{xy}, f_n^{xz}, f_n^{yz}\}$; (3) *Instruction Generation* decodes each part latent by leveraging its projection features to generate instruction parameters $\{(c_n^{\mathrm{skt}}, c_n^{\mathrm{ext}})\}_{n=1}^N$.

Given the input point cloud $\mathcal{X}$, our goal is to generate a sequence of structured parametric instructions $\mathcal{C} = \{c_n\}_{n=1}^N$, where each sketch–extrusion pair compactly describes a constructive step in the modeling process. Formally, each instruction $c_n = (c_n^{\mathrm{skt}}, c_n^{\mathrm{ext}})$ consists of two components: $c_n^{\mathrm{skt}}$ encodes 2D primitives (line, arc, and circle in our work) to define the profile geometry of the sketch, while $c_n^{\mathrm{ext}}$ specifies the corresponding solid operation via parameters such as orientation, origin, extrusion distance, scaling, and Boolean operation type. Following [42, 43, 48], both components are represented as discrete intervals obtained by quantizing continuous modeling parameters into fixed vocabularies. More details about CAD sequence representation are provided in Appendix A.1.

Inspired by the compositional nature of human CAD modeling, we introduce **PartCAD**, a semi-autoregressive framework that translates unstructured point clouds into structured parametric instructions. Instead of treating instruction generation as a flat sequence prediction task, PartCAD decomposes the input geometry into part-wise latent representations, each grounded in implicit design intent. Given that design details are inherently embedded within canonical planes and different perspective views encode distinct geometric and topological features, we extract triplane projection features to provide low-level geometric guidance for each part latent. Finally, each part latent is

decoded leveraging local- and global-level features, enabling accurate CAD instruction generation in a single forward pass. The rest of this section elaborates on these key techniques along with optimization strategies. Further implementation details are provided in Appendix A.2.

**Autoregressive Implicit Part Decomposition:**
Real-world CAD models are inherently compositional, comprising multiple parts with explicit semantics and ordered construction logic. To model this structure, we propose an autoregressive formulation that incrementally decomposes the input geometry into semantically meaningful latent representations. Given a global feature $f_{\mathcal{X}}^{\text{pc3d}}$ extracted from the input point cloud $\mathcal{X}$, the part decoder is initialized with a special start embedding $z_s$ and sequentially generates a series of implicit part latents $\{z_n\}_{n=1}^{N}$, each conditioned on the previously decoded latents and the shared global context. As shown in Figure 3, each latent acts as a semantic anchor for generating a corresponding modeling instruction. Formally, the generation process follows:

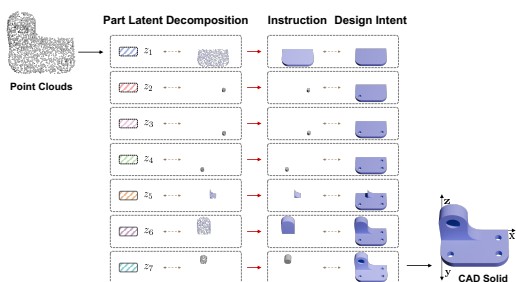

Figure 3: Given a 3D point cloud, PartCAD predicts a sequence of implicit part latent $z_n$, each encodes a part-level decomposition and is aligned with a modeling instruction that conveys explicit design intent.

$$P(\mathcal{Z} \mid f_{\mathcal{X}}^{\text{pc3d}}) = \prod_{n=1}^{N} P(z_n \mid z_{<n}, f_{\mathcal{X}}^{\text{pc3d}}) \tag{1}$$

To regulate this autoregressive process, we introduce a part discriminator that classifies whether each generated latent corresponds to a valid modeling operation. The discriminator provides auxiliary supervision to distinguish valid construction steps from redundant slots, enhancing instruction alignment and termination control.

**Triplane Projection Guidance:** Inspired by the observation that human CAD modeling often sketches in canonical planes and visualizes extrusions from side views, we propose a projection-guided approach that converts 3D geometry into structured 2D features for instruction decoding. These triplane projections capture view-specific design intent and serve as an effective interface between raw point clouds and parametric instruction synthesis. To this end, we estimate an orientation from each part latent $z_n$, rotate the input point cloud accordingly, and project it onto three canonical planes to extract aligned 2D features.

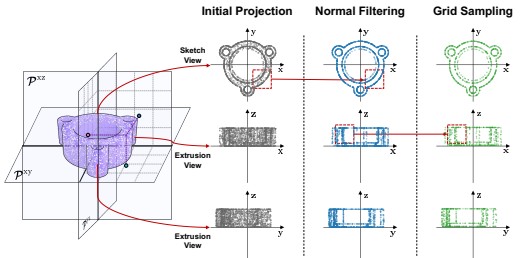

Figure 4: Illustration of adaptive projection. From left to right: initial projection, after normal filtering, and final result with adaptive grid sampling.

However, direct projections often introduce artifacts, such as points clustering along extrusion faces and sparse coverage in curved regions. To mitigate this, we propose an adaptive projection strategy that combines *normal-based filtering* and *adaptive grid sampling*. As illustrated in Figure 4, the former removes points whose surface normals deviate from the projection direction (thresholded by $\delta_{\text{normal}}$), while the latter resamples the filtered points into uniform grids, enforcing a spacing constraint $\delta_{\text{grid}}$ to ensure balanced coverage and reduce redundancy.

To effectively capture structural cues from projection points, we design a hierarchical KNN aggregation kernel tailored for triplane projections. After projection, geometrically distinct 3D regions may collapse into adjacent 2D neighborhoods, making naive Euclidean KNN prone to ambiguity. To address this, our kernel progressively refines $k$ neighbor selection in three stages: (1) select the nearest $3k$ candidates by 2D Euclidean distance; (2) filter to $2k$ points with similar radial distances from the projection centroid; and (3) retain the final $k$ neighbors with the most similar surface normals to enforce consistency. This results in a geometry-aware local neighborhood that preserves surface semantics for downstream feature extraction. Formally, the final neighborhood index is defined as:

$$\text{idx}_{\text{final}} = \text{TopK}_k\big(\|\mathbf{n}_i - \mathbf{n}_j\|,\ j \in \text{TopK}_{2k}(|r_i - r_j|,\ j \in \text{TopK}_{3k}(\|\mathbf{p}_i - \mathbf{p}_j\|))\big), \tag{2}$$

where $\mathbf{p}_i$ denotes the 2D projected coordinates, $\mathbf{n}_i$ the surface normal, and $r_i$ the radial distance to the centroid. After constructing the refined neighborhoods, projection features are extracted using three EdgeConv-style layers. Empirically, our hierarchical kernel enables more accurate encoding of sketch and extrusion structures. See Appendix B for additional comparisons and visualizations.

**CAD Instruction Generation:** Given the part-level representations from autoregressive decomposition, our goal is to decode complete parametric instructions in a single forward pass. As shown in Figure 2 (bottom right), we leverage the semantics encoded in each part latent $z_n$, as well as the geometric cues preserved in its corresponding triplane projections $f_n^{\mathrm{pc2d}} = \{f_n^{xy}, f_n^{xz}, f_n^{yz}\}$. Specifically, we employ two dedicated decoder branches: the top-view feature $f_n^{xy}$ is used for sketch decoder $\mathbb{D}_{\mathrm{skt}}$, while the side-view features $f_n^{xz}$ and $f_n^{yz}$ are concatenated to guide the extrusion decoder $\mathbb{D}_{\mathrm{ext}}$. Mathematically, the generation process can be formulated as:

$$c_n^{\mathrm{skt}} = \mathbb{D}_{\mathrm{skt}}(z_n, \ f_n^{xy}), \quad c_n^{\mathrm{ext}} = \mathbb{D}_{\mathrm{ext}}(z_n, \ [f_n^{xz}; \ f_n^{yz}]) \tag{3}$$

To facilitate instruction decoding from part latents, we introduce two interaction mechanisms that connect semantic embeddings with features extracted from triplane projections. First, a cross-attention (CA) module aligns each part latent $z_n$ with its associated point-level projection features $\mathbf{f}_n \in \mathbb{R}^{N \times d}$ to enable local feature aggregation. Second, a self-attention (SA) operates on global features $\bar{\mathbf{f}}_n = \mathrm{MaxPool}(\mathbf{f}_n) \in \mathbb{R}^{1 \times d}$ to facilitate contextual interaction.

$$\mathrm{CA}(z_n, \mathbf{f}_n) = \mathrm{softmax}\big(z_n W_Q^{\mathrm{ca}}(\mathbf{f}_n W_K^{\mathrm{ca}})^\top / \sqrt{d_k}\big)(\mathbf{f}_n W_V^{\mathrm{ca}}), \tag{4}$$

$$\mathrm{SA}(\bar{\mathbf{f}}_n) = \mathrm{softmax}\big(\bar{\mathbf{f}}_n W_Q^{\mathrm{sa}}(\bar{\mathbf{f}}_n W_K^{\mathrm{sa}})^\top / \sqrt{d_k}\big)(\bar{\mathbf{f}}_n W_V^{\mathrm{sa}}), \tag{5}$$

where $W_Q^{\mathrm{ca}}, W_K^{\mathrm{ca}}, W_V^{\mathrm{ca}}$ and $W_Q^{\mathrm{sa}}, W_K^{\mathrm{sa}}, W_V^{\mathrm{sa}}$ denote the learnable projection matrices, $\mathrm{softmax}$ is applied row-wise across the attention scores, and $d_k$ is the feature dimension of each head. The output embeddings of the above interactions serve as inputs to the final decoding: the CA results provide the query $Q$, while the SA outputs act as the key and value $K/V$.

$$F_n^{(l)} \leftarrow \mathrm{LN}(\mathrm{FFN}(\mathrm{LN}(F_n^{(l-1)} + \mathrm{Dropout}(\mathrm{MHA}(F_n^{(l-1)}, K_n, V_n))))), \quad l = 1, 2 \tag{6}$$

Here, $F_n^{(0)} = Q_n$, MHA denotes multi-head attention, LN is the Layer Normalization, and FFN is the feed forward network. Finally, the output embeddings after the second decoder layer are fed into two separate linear heads to generate the complete instruction:

$$c_n = \big(\mathrm{MLP}_{\mathrm{sketch}}(F_n^{(2,\mathrm{sketch})}), \ \mathrm{MLP}_{\mathrm{extrusion}}(F_n^{(2,\mathrm{extrusion})})\big), \quad \mathcal{C} = \{c_n\}_{n=1}^N \tag{7}$$

**Optimization:** As shown in Figure 2, we employ a multi-objective loss to jointly supervise the training process. First, the *Parameter losses* ($\mathcal{L}_{\mathrm{skt}}$ and $\mathcal{L}_{\mathrm{ext}}$), defined as position-wise cross-entropy over discretized instruction tokens, supervise sketch and extrusion prediction at each modeling step. Second, the *Rotation loss* ($\mathcal{L}_{\mathrm{rot}}$), formulated as cross-entropy over discretized orientation tokens, encourages accurate estimation of canonical-aligned rotations for reliable triplane projection. Last, we propose a *Validity loss* ($\mathcal{L}_{\mathrm{val}}$), implemented as binary cross-entropy between predicted validity scores and one-hot labels, to guide the part decoder in distinguishing meaningful construction steps. These labels indicate whether each sequence in the ground-truth instruction corresponds to a valid modeling operation. The overall training objective is the weighted sum of the above components:

$$\mathcal{L}_{\mathrm{total}} = \lambda_{\mathrm{skt}}\mathcal{L}_{\mathrm{skt}} + \lambda_{\mathrm{ext}}\mathcal{L}_{\mathrm{ext}} + \lambda_{\mathrm{rot}}\mathcal{L}_{\mathrm{rot}} + \lambda_{\mathrm{val}}\mathcal{L}_{\mathrm{val}}, \tag{8}$$

where $\lambda_{\mathrm{skt}}, \lambda_{\mathrm{ext}}, \lambda_{\mathrm{rot}}$ and $\lambda_{\mathrm{val}}$ are weight coefficients controlling the contribution of each term.

## 4   Experiments

**Dataset:** We train and evaluate our model on the DeepCAD dataset [42], and further perform cross-dataset validation on the Fusion 360 Gallery [12]. Both datasets provide executable modeling instructions that can be rendered into 3D geometry via standard CAD kernels. Following prior work [43], we remove duplicate data based on geometry similarities, yielding $\sim 140$k training and $\sim 7$k test/validation samples. We generate point clouds by uniformly sampling 2,048 points from the normalized CAD model. Instruction parameters are quantized to 8 bits, and each sequence is organized into *extrusion-sketch* pairs (see Appendix A.1 for details).

**Implementation and Training:** The latent dimension in PartCAD is set to 512. We use $k = 40$ neighbors for 3D point cloud encoding and $k = 60$ for projection feature extraction. For adaptive point cloud projection, we set the normal threshold $\delta_{\mathrm{normal}} = 0.5$ and grid spacing $\delta_{\mathrm{grid}} = 1 \times 10^{-6}$. The model is trained for 200 epochs with a batch size of 32 using the AdamW optimizer [54] (initial learning rate $1 \times 10^{-4}$) and an ExponentialLR scheduler. Loss weights are set as $\lambda_{\mathrm{skt}} = 2$, $\lambda_{\mathrm{ext}} = 1$, $\lambda_{\mathrm{val}} = 5$, and $\lambda_{\mathrm{rot}} = 1$. All experiments are conducted on 8 NVIDIA A100-40GB GPUs, with each training run taking around 18 hours. During inference, instructions are generated by top-1 decoding over the autoregressively predicted part latents. Additional details are provided in Appendix A.2.

**Evaluation metrics:** To comprehensively evaluate the quality of the generated CAD sequences and corresponding 3D shapes, we adopt three primary metrics: *Chamfer Distance* (CD), *F1 Score*, and *Invalid Rate* (IR). CD measures geometric similarity as the bidirectional distance between predicted and ground-truth shapes, each uniformly sampled with 8,192 points (scaled by $10^3$ for readability). We report both mean and median CD to reflect overall accuracy and robustness. Following [49, 48], we compute F1 scores for sketch and extrusion parameters to evaluate instruction-level consistency using the Hungarian algorithm [55]. Moreover, the ratio of predictions that fail to generate valid geometry using PythonOCC [10] is reported as the IR.

**Baselines:** We comprehensively compare our method with several representative CAD generation approaches, categorized into *sequence generation-based* [42, 43] and *primitive fitting-based* [15, 40, 14, 13] methods. For sequence-based baselines, we evaluate the generated construction instructions directly. For primitive-based methods, which predict geometry without explicit command sequences, we assess performance using CD and IR to measure geometric fidelity and modeling validity. All baselines are trained, fine-tuned, and tested using their official implementations, ensuring fair comparison against strong existing methods. Full implementation details are provided in the Appendix A.3.

## 4.1 Evaluation Results

**Design Instruction Generation Results:** Table 1 reports the quantitative results of CAD instruction generation on the DeepCAD dataset. The experiments demonstrate that our method outperforms all sequence generation-based methods across all metrics. For instruction parameter indicators like F1 and IR, the results show that our method achieves more accurate parameter recovery with minimal sequence-level errors. For geometric metrics such as median and mean CD, the results show that our method yields more accurate shape reconstruction with higher fidelity to the ground-truth geometry.

Table 1: Quantitative results of sequence generation-based methods on the DeepCAD dataset [42]. We report F1 scores for sketch parameters (Line, Arc, Circle) and extrusion parameters (Extrusion), CD (mean and median) for geometric fidelity, and IR for modeling validity.

| Methods | F1 Score | | | | CD ($\times 10^3$) | | IR $\downarrow$ |
| | Line $\uparrow$ | Arc $\uparrow$ | Circle $\uparrow$ | Extrusion $\uparrow$ | Mean $\downarrow$ | Median $\downarrow$ | |
| --- | --- | --- | --- | --- | --- | --- | --- |
| DeepCAD [42] | 69.37 | 15.45 | 60.26 | 87.56 | 44.53 | 17.49 | 8.51 |
| DeepCAD* [42] | 71.54 | 18.50 | 58.13 | 87.88 | 39.65 | 8.33 | 5.39 |
| TransCAD [43] | 75.03 | 40.52 | 73.89 | 89.17 | 34.54 | 5.67 | 3.60 |
| PartCAD (Ours) | **82.82** | **56.06** | **81.06** | **95.56** | **4.93** | **0.238** | **0.91** |

We also compare our method with primitive fitting-based approaches in terms of geometric fidelity and modeling validity. As reported in Table 2, our method achieves a lower median CD and IR, indicating more accurate geometry and fewer invalid predictions. The higher F1 scores indicate accurate recovery of instruction parameters, while lower CD reflects that the generated CAD models are geometrically consistent with the input point clouds. Figure 5 presents qualitative comparisons with all baselines. As illustrated, our method shows accurate and struc-

Table 2: Quantitative results of primitive fitting-based methods on the DeepCAD dataset [42] with median CD and IR metrics.

| Methods | Median CD ($\times 10^3$) $\downarrow$ | IR $\downarrow$ |
| --- | --- | --- |
| Point2Cyl [15] | 0.493 | 3.72 |
| ExtrudeNet [14] | 0.371 | 24.51 |
| SECAD [40] | 0.355 | 7.92 |
| HNC-CAD [13] | 0.841 | 6.94 |
| PartCAD (Ours) | **0.238** | **0.91** |

turally consistent reconstructions even on challenging cases involving complex sketches and multi-part assemblies. In contrast, sequence generation baselines (DeepCAD, DeepCAD*, TransCAD) fail to

generate valid instructions or produce inconsistent geometries in some examples, while primitive fitting-based methods (HNC-CAD, SECAD, Point2Cyl, ExtrudeNet) struggle to preserve fine-grained details. More visual results are provided in Figure 1 and Appendix C.1.

**Cross Dataset Experiments:** Following the protocol of prior works [39, 48, 43], we conduct the cross-dataset evaluation on Fusion 360 Gallery [12] to assess generalization performance. Results presented in Table 3 show that PartCAD significantly outperforms all sequence generation-based methods in both median CD and IR, reducing the median CD by an order of magnitude compared to baselines. While the median CD is slightly higher than that of some primitive fitting-based methods, which benefit from direct primitive-level alignment, our method can provide parametric design history with high validity, enabling more interpretable and editable CAD modeling. Representative results are shown in Figure 6, where our method reconstructs more complete and faithful CAD models compared to all baselines, better capturing the underlying geometry from input point clouds.

Table 3: Quantitative results of cross-dataset experiments on the Fusion 360 Gallery [12], measured in terms of median CD and IR.

| Methods | Median CD ($\times 10^3$) $\downarrow$ | IR $\downarrow$ |
|---|---|---|
| *Methods based on Primitive Fitting* | | |
| Point2Cyl [15] | 0.458 | 3.28 |
| ExtrudeNet [14] | 0.513 | 23.85 |
| SECAD [40] | **0.433** | 7.49 |
| HNC-CAD [13] | 3.785 | 6.83 |
| *Methods based on Sequence Generation* | | |
| DeepCAD [42] | 95.88 | 26.28 |
| DeepCAD* [42] | 44.23 | 6.90 |
| TransCAD [43] | 39.24 | 4.92 |
| PartCAD (Ours) | 1.130 | **1.31** |

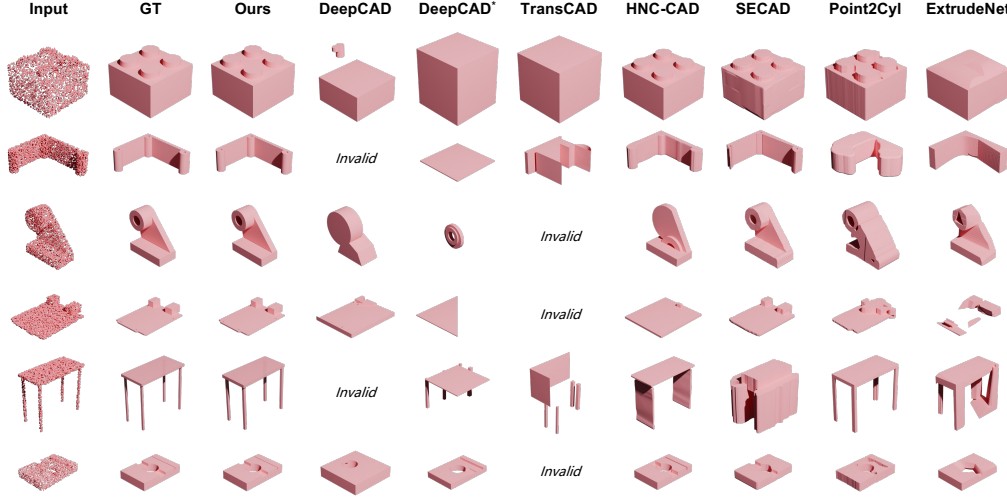

Figure 5: Qualitative results on the DeepCAD dataset [42]. Each row shows input point clouds (left) and the reconstructed CAD models by different methods (right). Compared with baseline methods, PartCAD produces geometrically accurate and structurally coherent shapes from the input point cloud, effectively preserving fine-grained details and part-wise layout.

**Real Scan Experiments:** To evaluate PartCAD in real-world applications, we 3D-printed several CAD models and captured their scans using a SIMSCAN-42 3D laser scanner. As shown in Figure 7, PartCAD successfully reconstructs geometrically faithful CAD models from the scanned point clouds. Furthermore, to evaluate

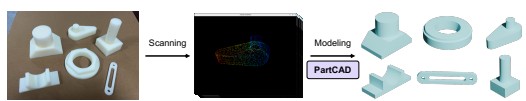

Figure 7: Experiments on real-life scanned point clouds (middle) show our method accurately reconstructs real-world CAD models.

the robustness of PartCAD under imperfect input conditions, we conduct additional experiments to assess the impact of degraded point cloud quality, including surface perturbations (e.g., noise contamination) and partial inputs (e.g., missing regions or occlusions). Detailed settings and qualitative visualizations are provided in Appendix C.2.

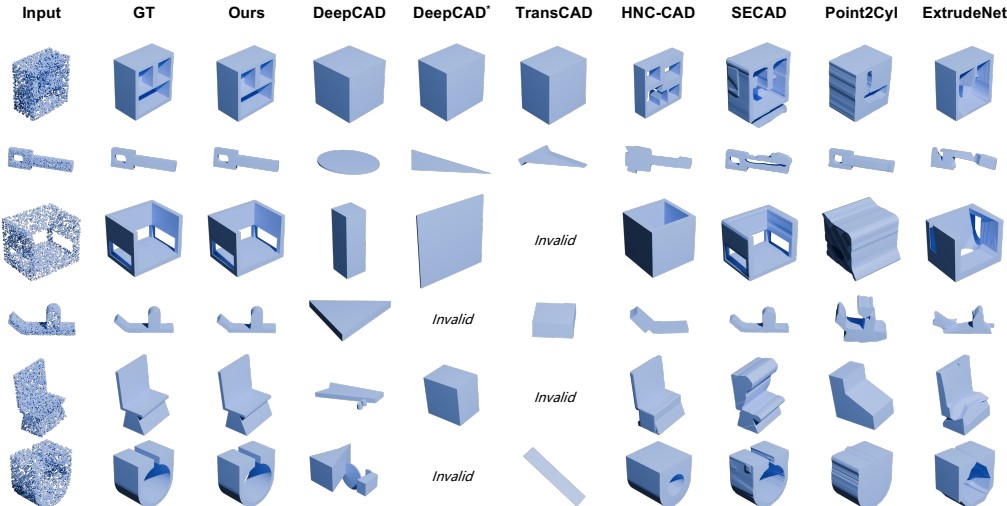

Figure 6: Qualitative results on the Fusion 360 Gallery [12]. Our approach produces structurally consistent CAD reconstructions with fine-grained geometric details. In contrast, DeepCAD* and TransCAD occasionally fail to generate valid models, while HNC-CAD, SECAD, Point2Cyl, and ExtrudeNet mainly recover coarse outlines but struggle to capture precise design features, often resulting in shape distortions or noisy artifacts.

## 4.2 Ablation Study

To analyze the contribution of each core component in the proposed PartCAD framework, we conduct ablations by selectively disabling key modules and report the results in Table 4. In the part-awareness branch, we remove the autoregressive part decoder and discriminator (*w/o Part Decomp.*), forcing direct orientation prediction from the global point cloud feature. This ablation results in severe degradation across all metrics, including a marked drop in F1 scores, increased mean and median CD, and a higher IR. A similar trend is observed when discarding triplane projection guidance (*w/o Proj. Guidance*) and relying solely on the global point cloud features. Notably, although retaining projection guidance improves the validity of generated instructions (declining IR), disabling adaptive sampling (*w/o Adapt. Proj.*) or replacing the hierarchical KNN kernel with standard Euclidean KNN (*w/o Hier. KNN*) leads to noticeable drops in parametric performance (F1) and geometric fidelity (CD). This indicates that these two components are essential for projection guidance to provide effective geometric cues. Finally, in the instruction decoding branch, we remove local feature aggregation (*w/o Local Feat. Aggreg.*) or disable global feature interaction (*w/o Global Feat. Interact.*). It can be noted that omitting either attention mechanism leads to higher CD and lower F1 scores, underscoring the necessity of both local and global information for accurate CAD instruction generation. Visual results of the ablation experiments can be found in Appendix C.3.

Table 4: Ablation study results on the DeepCAD dataset [42]. We report F1 scores for sketch (weighted average over Line, Arc, Circle) and extrusion parameters, Chamfer Distance (CD) for geometric fidelity, and Invalid Rate (IR) for instruction-level correctness.

| Group | Model Variant | F1 Score ↑ | | CD ($\times 10^3$) ↓ | | IR ↓ |
| --- | --- | --- | --- | --- | --- | --- |
| | | Sketch | Extrusion | Mean | Median | |
| Part Awareness | w/o Part Decomp. | 59.51 | 72.63 | 15.34 | 1.987 | 8.01 |
| Projection Guidance | w/o Proj. Guidance | 65.53 | 87.23 | 14.59 | 1.062 | 8.94 |
| | w/o Adapt. Proj. | 61.33 | 73.46 | 13.42 | 1.282 | 5.79 |
| | w/o Hier. KNN | 64.71 | 78.93 | 10.27 | 1.384 | 4.27 |
| Instruction Generation | w/o Local Feat. Aggreg. | 75.58 | 95.25 | 5.639 | 0.313 | 1.33 |
| | w/o Global Feat. Interact. | 73.61 | 94.63 | 6.301 | 0.482 | 1.51 |
| **PartCAD (Full Model)** | | **80.01** | **95.56** | **4.93** | **0.238** | **0.91** |

## 5  Discussions

**Conclusion:** We have introduced PartCAD, a semi-autoregressive framework designed for generating structured CAD modeling instructions from point clouds. Our contributions include an implicit part decomposition strategy that autoregressively derives interpretable part representations from point clouds, a projection-guidance module that provides explicit geometric cues via triplane features, and a non-autoregressive decoder that generates parametric instructions by fusing point- and part-wise features. Extensive evaluations show that PartCAD outperforms existing baselines, achieving significant improvements in both parametric accuracy and geometric fidelity.

**Limitations and Future Work:** There are several limitations in our current work. First, the performance of the model is inherently constrained by the size and diversity of the training datasets. Second, the mapping from geometry to parametric sequence can be intrinsically one-to-many, introducing ambiguity that complicates both training and evaluation. Third, generating accurate modeling instructions becomes increasingly challenging as object complexity grows, particularly for geometrically intricate or hierarchically unstructured models. Some failure cases are discussed in Appendix D. In future, we plan to delve into more advanced reasoning mechanisms and richer representations to resolve the ambiguity, optimize the instruction generation, and support intricate shape designs, thereby enhancing the robustness and applicability of PartCAD.

**Broader Impact:** While this work has the potential to support educational tools for engineering and design students and lower the barrier to entry for CAD design, its focus on sketch-extrusion alone may constrain design creativity.

## 6  Acknowledgments

This work was supported in part by the National Natural Science Foundation of China (NSFC) under the Distinguished Young Scholars Program (Grant 62025301), the NSFC Basic Science Center Program (Grant 62088101), the MOE AcRF Tier 1 Grant of Singapore (RG12/22), and the China Scholarship Council (Grant 202406030027).

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

# NeurIPS Paper Checklist

1. **Claims**

   Question: Do the main claims made in the abstract and introduction accurately reflect the paper's contributions and scope?

   Answer: [Yes]

   Justification: The abstract and introduction clearly state the central contributions of the paper: a semi-autoregressive framework for CAD instruction generation that decomposes input geometry into part-wise latent representations and decodes them using triplane projection guidance. These claims are consistently supported by detailed methodological descriptions and validated through both quantitative and qualitative experiments. The scope and limitations of the approach are appropriately discussed, and no unsubstantiated or overgeneralized claims are made. The presentation remains faithful to what is actually achieved in the work.

   Guidelines:

   - The answer NA means that the abstract and introduction do not include the claims made in the paper.
   - The abstract and/or introduction should clearly state the claims made, including the contributions made in the paper and important assumptions and limitations. A No or NA answer to this question will not be perceived well by the reviewers.
   - The claims made should match theoretical and experimental results, and reflect how much the results can be expected to generalize to other settings.
   - It is fine to include aspirational goals as motivation as long as it is clear that these goals are not attained by the paper.

2. **Limitations**

   Question: Does the paper discuss the limitations of the work performed by the authors?

   Answer: [Yes]

   Justification: The paper discusses key limitations in Section 5 and Appendix D. First, the performance of the proposed model is bounded by the scale and diversity of available training data. Second, the inherently one-to-many nature of geometry-to-instruction mapping introduces ambiguity that complicates both learning and evaluation. Third, the model's ability to generate accurate CAD instructions declines for complex or hierarchically unstructured geometries. We provide representative failure cases and discuss how these limitations affect generalization and robustness. Additionally, we outline future directions to address these issues through improved reasoning mechanisms and richer representations.

   Guidelines:

   - The answer NA means that the paper has no limitation while the answer No means that the paper has limitations, but those are not discussed in the paper.
   - The authors are encouraged to create a separate "Limitations" section in their paper.
   - The paper should point out any strong assumptions and how robust the results are to violations of these assumptions (e.g., independence assumptions, noiseless settings, model well-specification, asymptotic approximations only holding locally). The authors should reflect on how these assumptions might be violated in practice and what the implications would be.
   - The authors should reflect on the scope of the claims made, e.g., if the approach was only tested on a few datasets or with a few runs. In general, empirical results often depend on implicit assumptions, which should be articulated.
   - The authors should reflect on the factors that influence the performance of the approach. For example, a facial recognition algorithm may perform poorly when image resolution is low or images are taken in low lighting. Or a speech-to-text system might not be used reliably to provide closed captions for online lectures because it fails to handle technical jargon.
   - The authors should discuss the computational efficiency of the proposed algorithms and how they scale with dataset size.

- If applicable, the authors should discuss possible limitations of their approach to address problems of privacy and fairness.
- While the authors might fear that complete honesty about limitations might be used by reviewers as grounds for rejection, a worse outcome might be that reviewers discover limitations that aren't acknowledged in the paper. The authors should use their best judgment and recognize that individual actions in favor of transparency play an important role in developing norms that preserve the integrity of the community. Reviewers will be specifically instructed to not penalize honesty concerning limitations.

3. **Theory assumptions and proofs**

Question: For each theoretical result, does the paper provide the full set of assumptions and a complete (and correct) proof?

Answer: [NA]

Justification: The paper does not contain theoretical results requiring formal assumptions or proofs. The work focuses on algorithmic design, architectural innovations, and empirical evaluation.

Guidelines:

- The answer NA means that the paper does not include theoretical results.
- All the theorems, formulas, and proofs in the paper should be numbered and cross-referenced.
- All assumptions should be clearly stated or referenced in the statement of any theorems.
- The proofs can either appear in the main paper or the supplemental material, but if they appear in the supplemental material, the authors are encouraged to provide a short proof sketch to provide intuition.
- Inversely, any informal proof provided in the core of the paper should be complemented by formal proofs provided in appendix or supplemental material.
- Theorems and Lemmas that the proof relies upon should be properly referenced.

4. **Experimental result reproducibility**

Question: Does the paper fully disclose all the information needed to reproduce the main experimental results of the paper to the extent that it affects the main claims and/or conclusions of the paper (regardless of whether the code and data are provided or not)?

Answer: [Yes]

Justification: The paper provides comprehensive implementation details, including architecture configurations, training hyperparameters, loss formulations, and dataset processing steps. Key components such as autoregressive part decomposition, triplane feature guidance, and CAD instruction generation are fully described in Section 3, Section 4, and Appendix A.2. All experimental protocols follow standard benchmarks with clearly stated evaluation metrics. These details are sufficient to reproduce the main results even without access to the source code.

Guidelines:

- The answer NA means that the paper does not include experiments.
- If the paper includes experiments, a No answer to this question will not be perceived well by the reviewers: Making the paper reproducible is important, regardless of whether the code and data are provided or not.
- If the contribution is a dataset and/or model, the authors should describe the steps taken to make their results reproducible or verifiable.
- Depending on the contribution, reproducibility can be accomplished in various ways. For example, if the contribution is a novel architecture, describing the architecture fully might suffice, or if the contribution is a specific model and empirical evaluation, it may be necessary to either make it possible for others to replicate the model with the same dataset, or provide access to the model. In general. releasing code and data is often one good way to accomplish this, but reproducibility can also be provided via detailed instructions for how to replicate the results, access to a hosted model (e.g., in the case of a large language model), releasing of a model checkpoint, or other means that are appropriate to the research performed.

- While NeurIPS does not require releasing code, the conference does require all submissions to provide some reasonable avenue for reproducibility, which may depend on the nature of the contribution. For example
    (a) If the contribution is primarily a new algorithm, the paper should make it clear how to reproduce that algorithm.
    (b) If the contribution is primarily a new model architecture, the paper should describe the architecture clearly and fully.
    (c) If the contribution is a new model (e.g., a large language model), then there should either be a way to access this model for reproducing the results or a way to reproduce the model (e.g., with an open-source dataset or instructions for how to construct the dataset).
    (d) We recognize that reproducibility may be tricky in some cases, in which case authors are welcome to describe the particular way they provide for reproducibility. In the case of closed-source models, it may be that access to the model is limited in some way (e.g., to registered users), but it should be possible for other researchers to have some path to reproducing or verifying the results.

5. **Open access to data and code**

    Question: Does the paper provide open access to the data and code, with sufficient instructions to faithfully reproduce the main experimental results, as described in supplemental material?

    Answer: [Yes]

    Justification: All implementation and training details are described in the main paper and Appendix to support reproducibility.

    Guidelines:

    - The answer NA means that paper does not include experiments requiring code.
    - Please see the NeurIPS code and data submission guidelines (`https://nips.cc/public/guides/CodeSubmissionPolicy`) for more details.
    - While we encourage the release of code and data, we understand that this might not be possible, so "No" is an acceptable answer. Papers cannot be rejected simply for not including code, unless this is central to the contribution (e.g., for a new open-source benchmark).
    - The instructions should contain the exact command and environment needed to run to reproduce the results. See the NeurIPS code and data submission guidelines (`https://nips.cc/public/guides/CodeSubmissionPolicy`) for more details.
    - The authors should provide instructions on data access and preparation, including how to access the raw data, preprocessed data, intermediate data, and generated data, etc.
    - The authors should provide scripts to reproduce all experimental results for the new proposed method and baselines. If only a subset of experiments are reproducible, they should state which ones are omitted from the script and why.
    - At submission time, to preserve anonymity, the authors should release anonymized versions (if applicable).
    - Providing as much information as possible in supplemental material (appended to the paper) is recommended, but including URLs to data and code is permitted.

6. **Experimental setting/details**

    Question: Does the paper specify all the training and test details (e.g., data splits, hyperparameters, how they were chosen, type of optimizer, etc.) necessary to understand the results?

    Answer: [Yes]

    Justification: We provide comprehensive details of our experimental settings, including dataset preprocessing, training/test splits, model architecture and hyperparameters, optimizer configurations, and loss weight settings in Section 4 and Appendix A.2. All these details are fully documented to ensure the results are understandable and reproducible.

    Guidelines:

- The answer NA means that the paper does not include experiments.
- The experimental setting should be presented in the core of the paper to a level of detail that is necessary to appreciate the results and make sense of them.
- The full details can be provided either with the code, in appendix, or as supplemental material.

7. **Experiment statistical significance**

Question: Does the paper report error bars suitably and correctly defined or other appropriate information about the statistical significance of the experiments?

Answer: [Yes]

Justification: All reported results are obtained through multiple carefully controlled runs with consistent data splits and training configurations. The results are highly stable, and consistent trends are observed across all baselines and experimental settings. These outcomes provide strong support for the validity of our claims.

Guidelines:

- The answer NA means that the paper does not include experiments.
- The authors should answer "Yes" if the results are accompanied by error bars, confidence intervals, or statistical significance tests, at least for the experiments that support the main claims of the paper.
- The factors of variability that the error bars are capturing should be clearly stated (for example, train/test split, initialization, random drawing of some parameter, or overall run with given experimental conditions).
- The method for calculating the error bars should be explained (closed form formula, call to a library function, bootstrap, etc.)
- The assumptions made should be given (e.g., Normally distributed errors).
- It should be clear whether the error bar is the standard deviation or the standard error of the mean.
- It is OK to report 1-sigma error bars, but one should state it. The authors should preferably report a 2-sigma error bar than state that they have a 96% CI, if the hypothesis of Normality of errors is not verified.
- For asymmetric distributions, the authors should be careful not to show in tables or figures symmetric error bars that would yield results that are out of range (e.g. negative error rates).
- If error bars are reported in tables or plots, The authors should explain in the text how they were calculated and reference the corresponding figures or tables in the text.

8. **Experiments compute resources**

Question: For each experiment, does the paper provide sufficient information on the computer resources (type of compute workers, memory, time of execution) needed to reproduce the experiments?

Answer: [Yes]

Justification: All experiments were conducted on 8 NVIDIA A100-PCIE-40GB GPUs, using a batch size of 32 and training for 200 epochs. The average training time of the full model was approximately 18 hours. All major experiments, including ablation studies and cross-dataset evaluations, were performed under this setup. Detailed implementation and resource settings are provided in Section 4 and Appendix A.2 to ensure reproducibility and transparency.

Guidelines:

- The answer NA means that the paper does not include experiments.
- The paper should indicate the type of compute workers CPU or GPU, internal cluster, or cloud provider, including relevant memory and storage.
- The paper should provide the amount of compute required for each of the individual experimental runs as well as estimate the total compute.
- The paper should disclose whether the full research project required more compute than the experiments reported in the paper (e.g., preliminary or failed experiments that didn't make it into the paper).

9. **Code of ethics**

Question: Does the research conducted in the paper conform, in every respect, with the NeurIPS Code of Ethics https://neurips.cc/public/EthicsGuidelines?

Answer: [Yes]

Justification: This research fully adheres to the NeurIPS Code of Ethics. It involves no human subjects, personal data, or sensitive content and does not pose foreseeable risks related to safety, privacy, fairness, or misuse. All experiments are conducted using publicly available datasets with appropriate licenses. We have carefully reviewed the ethical guidelines and confirm that our study complies with all relevant principles.

Guidelines:

- The answer NA means that the authors have not reviewed the NeurIPS Code of Ethics.
- If the authors answer No, they should explain the special circumstances that require a deviation from the Code of Ethics.
- The authors should make sure to preserve anonymity (e.g., if there is a special consideration due to laws or regulations in their jurisdiction).

10. **Broader impacts**

Question: Does the paper discuss both potential positive societal impacts and negative societal impacts of the work performed?

Answer: [Yes]

Justification: Our work has the potential to support educational tools for engineering and design students by automating CAD modeling and lowering the barrier to entry for novice users. This may improve accessibility and efficiency in industrial design workflows, particularly in manufacturing, architecture, and engineering, by reducing design time and enabling more interpretable modeling tools. However, as our method centers on sketch–extrusion operations, it may constrain modeling flexibility in more complex or creative design scenarios. Moreover, while the method itself does not pose direct societal risks, we recognize the potential for misuse in automated design pipelines, such as the unauthorized replication of copyrighted or proprietary structures. To mitigate such concerns, we advocate for ethical deployment in compliance with intellectual property regulations and encourage future work to expand the modeling expressiveness and reinforce responsible use in educational and professional contexts.

Guidelines:

- The answer NA means that there is no societal impact of the work performed.
- If the authors answer NA or No, they should explain why their work has no societal impact or why the paper does not address societal impact.
- Examples of negative societal impacts include potential malicious or unintended uses (e.g., disinformation, generating fake profiles, surveillance), fairness considerations (e.g., deployment of technologies that could make decisions that unfairly impact specific groups), privacy considerations, and security considerations.
- The conference expects that many papers will be foundational research and not tied to particular applications, let alone deployments. However, if there is a direct path to any negative applications, the authors should point it out. For example, it is legitimate to point out that an improvement in the quality of generative models could be used to generate deepfakes for disinformation. On the other hand, it is not needed to point out that a generic algorithm for optimizing neural networks could enable people to train models that generate Deepfakes faster.
- The authors should consider possible harms that could arise when the technology is being used as intended and functioning correctly, harms that could arise when the technology is being used as intended but gives incorrect results, and harms following from (intentional or unintentional) misuse of the technology.
- If there are negative societal impacts, the authors could also discuss possible mitigation strategies (e.g., gated release of models, providing defenses in addition to attacks, mechanisms for monitoring misuse, mechanisms to monitor how a system learns from feedback over time, improving the efficiency and accessibility of ML).

11. **Safeguards**

Question: Does the paper describe safeguards that have been put in place for responsible release of data or models that have a high risk for misuse (e.g., pretrained language models, image generators, or scraped datasets)?

Answer: [NA]

Justification: Our work does not involve high-risk models or datasets with potential for misuse, such as large-scale generative models or scraped web data. The proposed method focuses on CAD instruction synthesis from point clouds, with no foreseeable dual-use or safety concerns.

Guidelines:

- The answer NA means that the paper poses no such risks.
- Released models that have a high risk for misuse or dual-use should be released with necessary safeguards to allow for controlled use of the model, for example by requiring that users adhere to usage guidelines or restrictions to access the model or implementing safety filters.
- Datasets that have been scraped from the Internet could pose safety risks. The authors should describe how they avoided releasing unsafe images.
- We recognize that providing effective safeguards is challenging, and many papers do not require this, but we encourage authors to take this into account and make a best faith effort.

12. **Licenses for existing assets**

Question: Are the creators or original owners of assets (e.g., code, data, models), used in the paper, properly credited and are the license and terms of use explicitly mentioned and properly respected?

Answer: [Yes]

Justification: All datasets and code assets used in this paper are publicly available and properly cited, including the DeepCAD dataset [42] and the Fusion 360 Gallery [12], both of which are distributed under open academic licenses. We strictly adhere to the terms of use and include appropriate attribution to the original sources in the paper. We also use publicly available codebases (e.g., PythonOCC [10]) that are released under permissive open-source licenses, all of which are appropriately credited in the manuscript.

Guidelines:

- The answer NA means that the paper does not use existing assets.
- The authors should cite the original paper that produced the code package or dataset.
- The authors should state which version of the asset is used and, if possible, include a URL.
- The name of the license (e.g., CC-BY 4.0) should be included for each asset.
- For scraped data from a particular source (e.g., website), the copyright and terms of service of that source should be provided.
- If assets are released, the license, copyright information, and terms of use in the package should be provided. For popular datasets, `paperswithcode.com/datasets` has curated licenses for some datasets. Their licensing guide can help determine the license of a dataset.
- For existing datasets that are re-packaged, both the original license and the license of the derived asset (if it has changed) should be provided.
- If this information is not available online, the authors are encouraged to reach out to the asset's creators.

13. **New assets**

Question: Are new assets introduced in the paper well documented and is the documentation provided alongside the assets?

Answer: [Yes]

Justification: We introduce a new trained CAD instruction generation model (PartCAD), including architecture details, training configuration, dependencies, and evaluation instructions. The documentation will be provided to ensure reproducibility and ease of use.

Guidelines:

- The answer NA means that the paper does not release new assets.
- Researchers should communicate the details of the dataset/code/model as part of their submissions via structured templates. This includes details about training, license, limitations, etc.
- The paper should discuss whether and how consent was obtained from people whose asset is used.
- At submission time, remember to anonymize your assets (if applicable). You can either create an anonymized URL or include an anonymized zip file.

14. **Crowdsourcing and research with human subjects**

Question: For crowdsourcing experiments and research with human subjects, does the paper include the full text of instructions given to participants and screenshots, if applicable, as well as details about compensation (if any)?

Answer: [NA]

Justification: This work does not involve any human subjects or crowdsourcing experiments.

Guidelines:

- The answer NA means that the paper does not involve crowdsourcing nor research with human subjects.
- Including this information in the supplemental material is fine, but if the main contribution of the paper involves human subjects, then as much detail as possible should be included in the main paper.
- According to the NeurIPS Code of Ethics, workers involved in data collection, curation, or other labor should be paid at least the minimum wage in the country of the data collector.

15. **Institutional review board (IRB) approvals or equivalent for research with human subjects**

Question: Does the paper describe potential risks incurred by study participants, whether such risks were disclosed to the subjects, and whether Institutional Review Board (IRB) approvals (or an equivalent approval/review based on the requirements of your country or institution) were obtained?

Answer: [NA]

Justification: This work does not involve any form of crowdsourcing or research with human subjects. All experiments are conducted using publicly available or synthetically generated datasets.

Guidelines:

- The answer NA means that the paper does not involve crowdsourcing nor research with human subjects.
- Depending on the country in which research is conducted, IRB approval (or equivalent) may be required for any human subjects research. If you obtained IRB approval, you should clearly state this in the paper.
- We recognize that the procedures for this may vary significantly between institutions and locations, and we expect authors to adhere to the NeurIPS Code of Ethics and the guidelines for their institution.
- For initial submissions, do not include any information that would break anonymity (if applicable), such as the institution conducting the review.

16. **Declaration of LLM usage**

Question: Does the paper describe the usage of LLMs if it is an important, original, or non-standard component of the core methods in this research? Note that if the LLM is used only for writing, editing, or formatting purposes and does not impact the core methodology, scientific rigorousness, or originality of the research, declaration is not required.

Answer: [NA]

Justification: This work does not involve any large language models (LLMs) as part of the core methodology. LLMs were not used in the development, implementation, or evaluation of the proposed approach.

Guidelines:

- The answer NA means that the core method development in this research does not involve LLMs as any important, original, or non-standard components.
- Please refer to our LLM policy (`https://neurips.cc/Conferences/2025/LLM`) for what should or should not be described.

# Appendix Overview

This appendix provides supplementary information to support the main paper. Section A presents detailed implementation settings for CAD sequence representation and the PartCAD framework. Section B offers additional methodological insights into adaptive projection refinement and hierarchical KNN aggregation. Section C reports additional experimental results and analyses. Section D provides some failure cases and more discussion.

# A    Additional Implementation Details

## A.1    CAD Representation

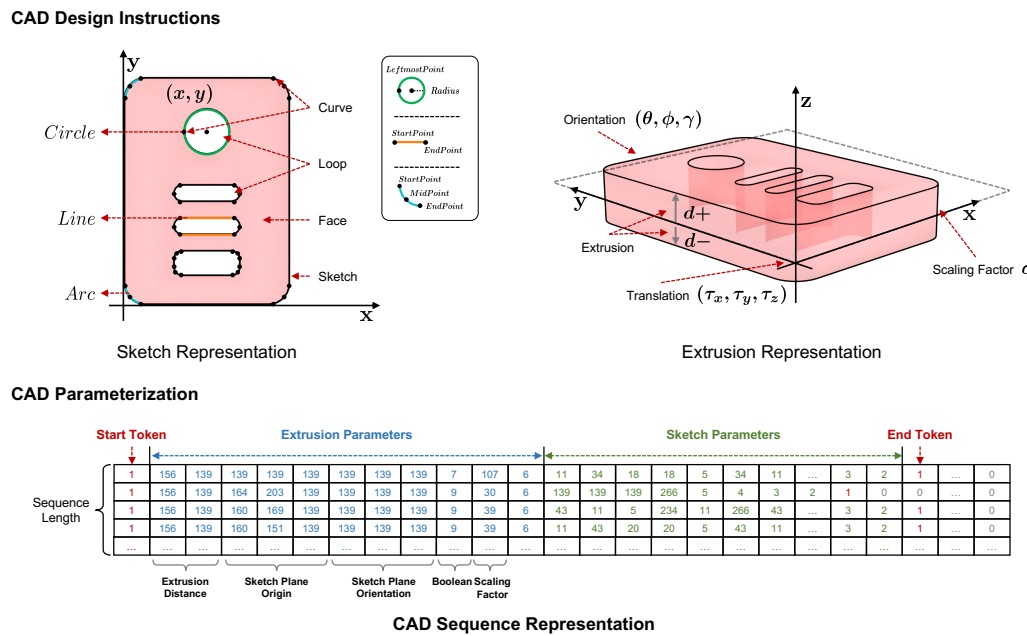

Figure 8: Illustration of CAD model and tokenized sequence representation.

We adopt a hierarchical CAD sequence representation structured as a sequence of *extrusion-sketch* token pairs. As illustrated in the bottom part of Figure 8, we formulate each sequence as a solid modeling instruction, consisting of extrusion parameters, sketch primitives, and start/end tokens. Compared to primitive-level [42] or tightly packed autoregressive formats [49, 48], our representation offers clearer semantic alignment with human design intent.

In the following, we detail the representation format and parameterization of each component.

**Sketch Representation:** As shown in the top-left part of Figure 8, each $\mathrm{Sketch}$ comprises one or more $\mathrm{Face}$, each bounded by one or more closed $\mathrm{Loop}$. Each loop consists of geometric primitives $\mathrm{Curve}$, i.e., *Line*, *Arc*, or *Circle*, which are arranged in counterclockwise order to ensure consistency. Each primitive is parameterized using several 2D coordinates $(x, y)$ (or radius $r$ for *Circle*):

- **Line:** Start Point and End Point
- **Arc:** Start Point, Midpoint, and End Point
- **Circle:** Leftmost Point and Radius

**Extrusion Representation:** As shown in the top-right part of Figure 8, each $\mathrm{Extrusion}$ instruction consists of 10 parameters that define how a 2D sketch is transformed into a 3D solid, including the $\mathrm{Orientation}$, $\mathrm{Translation}$, and $\mathrm{Scaling\ Factor}$ of the sketch plane, as well as the extrusion extent and the Boolean operation type:

- **Orientation:** 3 parameters $(\theta, \phi, \gamma)$ that define the sketch plane orientation via Euler angles
- **Translation:** 3 parameters $(\tau_x, \tau_y, \tau_z)$ specifying the sketch plane position
- **Scaling Factor:** 1 parameter $\sigma$ controlling the normalization of sketch size
- **Extrusion Extent:** 2 parameters $(d^+, d^-)$ indicating extrusion distances along and opposite to the sketch normal
- **Boolean Operation:** 1 parameter $\beta$ denoting the extrusion operation type, including *new*, *cut*, *join*, and *intersection*

**Tokenization and Quantization.** To enable structured sequence representation, we follow [48, 49] to introduce special tokens for parameter separation, including padding, start/end of sequence and end markers for curves, loops, faces, sketches, and extrusions. All continuous parameters (except special tokens and Boolean operation types) are uniformly quantized into 8-bit integers, as shown in Table 5.

Table 5: Token vocabulary in CAD sequence representation.

| Token Category | Token Symbol | Token Value | Description |
|---|---|---|---|
| Sketch Token | $x$ | $[\![11, \ldots, 266]\!]$ | X coordinate |
| | $y$ | $[\![11, \ldots, 266]\!]$ | Y coordinate |
| | $r$ | $[\![11, \ldots, 266]\!]$ | Circle Radius |
| Extrusion Token | $\theta$ | $[\![11, \ldots, 266]\!]$ | |
| | $\phi$ | $[\![11, \ldots, 266]\!]$ | Sketch Plane Orientation |
| | $\gamma$ | $[\![11, \ldots, 266]\!]$ | |
| | $\tau_x$ | $[\![11, \ldots, 266]\!]$ | |
| | $\tau_y$ | $[\![11, \ldots, 266]\!]$ | Sketch Plane Origin |
| | $\tau_z$ | $[\![11, \ldots, 266]\!]$ | |
| | $d^+$ | $[\![11, \ldots, 266]\!]$ | Extrusion Distance Toward and |
| | $d^-$ | $[\![11, \ldots, 266]\!]$ | Opposite Sketch Plane Normal |
| | $\sigma$ | $[\![11, \ldots, 266]\!]$ | Sketch Scaling Factor |
| | $\beta$ | $\{7, 8, 9, 10\}$ | Boolean (*New*, *Cut*, *Join*, *Intersect*) |
| Special Token | PAD | 0 | Padding token |
| | SOS | 1 | Start of sequence |
| | EOS | 1 | End of sequence |
| | $e_s$ | 2 | End of sketch |
| | $e_f$ | 3 | End of face |
| | $e_l$ | 4 | End of loop |
| | $e_c$ | 5 | End of curve |
| | $e_e$ | 6 | End of extrusion |

**Implementation Details.** We set the maximum sequence length, i.e., the maximum number of *extrusion-sketch* pairs per CAD model, to 10. Each sequence is padded to a fixed length of 110 tokens to ensure batching during training. Following [42], each face contains up to 6 loops, and each loop comprises at most 15 curve primitives. In addition, we follow [43] to perform deduplication based on geometric similarity. While the filtered dataset may still include models with similar shapes (e.g., cubes or cylinders), variations in size, position, and orientation ensure that no exact sequence duplicates remain in the final dataset.

## A.2 Implementation Details for PartCAD Architecture

The **PartCAD** framework consists of three modules: (1) *Autoregressive Implicit Part Decomposition*, (2) *Triplane Projection Guidance*, and (3) *CAD Instruction Generation*. Below, we detail the architectural configurations and hyperparameters for each component.

**Autoregressive Implicit Part Decomposition:** The 3D point cloud encoder is implemented using three stacked EdgeConv layers [22] with channel dimensions $[12, 64, 64, 128]$, followed by global

max pooling. The input feature consists of 3D coordinates and surface normals, and the neighborhood size is set to $k = 40$. The output global feature is projected via a linear projection head.

The autoregressive part decoder comprises four Transformer decoder layers [56], each with 8 attention heads and a feed-forward network of hidden dimension 2048. It incorporates residual connections, layer normalization, and a dropout rate of 0.2. For auxiliary supervision and orientation prediction, we append two prediction heads:

*Part Discriminator:* The part discriminator, implemented as a single-layer MLP of size $[512, 1]$, outputs a scalar confidence score indicating whether the generated latent corresponds to a valid modeling step.

*Orientation Predictor:* The orientation predictor, implemented as a single-layer MLP of size $[512, 3]$, predicts discretized sketch plane orientation tokens $(\theta, \phi, \gamma)$ used to align projections with canonical CAD views.

**Triplane Projection Guidance:** With the predicted orientation $(\theta_n, \phi_n, \gamma_n)$ from each part latent, we perform denumericalization to obtain a rotation matrix $\mathbf{R}_n = \text{EulerToMatrix}(\theta_n, \phi_n, \gamma_n)$. This is then applied to the input point cloud $\mathcal{X}$ to produce a rotated geometry $\mathcal{X}_n^{\text{rot}} = \mathbf{R}_n \cdot \mathcal{X}$, aligning it with canonical CAD design views.

Based on $\mathcal{X}_n^{\text{rot}}$, we derive canonical-aligned triplane projections for subsequent adaptive sampling and feature encoding. To enable reliable triplane encoding, we apply a projection refinement module using thresholds $\delta_{\text{normal}} = 0.5$ (for normal filtering) and $\delta_{\text{grid}} = 1 \times 10^{-6}$ (for grid resampling).

The 2D projection encoder consists of three 2D EdgeConv layers with dimensions $[12, 64, 64, 64]$. Input features include 2D projected coordinates, associated normals, and radial distances to the projection centroid. The local neighborhood size is set to $k = 60$. The encoder outputs per-point features, which are subsequently used for instruction decoding.

**CAD Instruction Generation:** We use separate decoding branches for sketch and extrusion parameter prediction. Each branch consists of: a *local feature aggregation layer* for aligning part latents with point-level projection features via cross-attention, a *global feature interaction layer* for contextual reasoning over triplane global features via self-attention, and two Transformer decoder layers (each with 8 attention heads, a feed-forward dimension of 2048, and a dropout rate of 0.2) for final sequence decoding. The output of each branch is passed through a linear layer to generate the predicted instruction parameters.

**Latent Dimensions:** The latent dimensions for the part representations $z_n$, 3D point features $f_{\mathcal{X}}^{\text{pc3d}}$, and $xy$-plane features $f_n^{xy}$ are all set to 512. The $xz$- and $yz$-plane features $f_n^{xz}$ and $f_n^{yz}$ are both set to 256 for dimensional compatibility during feature concatenation, i.e., $[f_n^{xz}; f_n^{yz}] \in \mathbb{R}^{512}$.

**Loss Implementation Details:** We implement the multi-objective loss described in Section 3, which combines supervision on instruction parameters with auxiliary objectives for orientation estimation and part validity classification.

*Parameter loss:* The parameter loss consists of two parts, i.e., the sketch loss $\mathcal{L}_{\text{skt}}$ and the extrusion loss $\mathcal{L}_{\text{ext}}$. Let $y_t$ denote the ground-truth token at position $t$ and $\hat{y}_t$ be the predicted logits, the loss is computed using position-wise cross-entropy:

$$\mathcal{L}_{\text{skt}} = -\sum_{t=1}^{T_{\text{skt}}} \sum_{i=1}^{d} y_{t,i}^{\text{skt}} \log \hat{y}_{t,i}^{\text{skt}}, \quad \mathcal{L}_{\text{ext}} = -\sum_{t=1}^{T_{\text{ext}}} \sum_{i=1}^{d} y_{t,i}^{\text{ext}} \log \hat{y}_{t,i}^{\text{ext}} \tag{9}$$

Here, $T_{\text{skt}}$ and $T_{\text{ext}}$ denote the sequence lengths of sketch and extrusion tokens respectively, $d = 267$ is the vocabulary size, and $y_{t,i}, \hat{y}_{t,i}$ denote the one-hot target and softmax prediction for token $t$ and class $i$.

*Rotation loss:* The rotation parameter is treated as a discrete token similar to other instruction parameters. Let $y_t^{\text{rot}}$ denote the one-hot target token and $\hat{y}_t^{\text{rot}}$ the predicted softmax probability at the corresponding rotation position, the loss is computed as:

$$\mathcal{L}_{\text{rot}} = -\sum_{i=1}^{d_{\text{rot}}} y_{t,i}^{\text{rot}} \log \hat{y}_{t,i}^{\text{rot}} \tag{10}$$

*Validity loss:* To supervise part validity, we define a binary label $y_n^{\mathrm{val}} \in \{0, 1\}$ for each part latent, indicating whether it corresponds to a valid modeling step. The prediction $\hat{y}_n^{\mathrm{val}} \in [0, 1]$ is obtained via a sigmoid function. The validity loss is computed using binary cross-entropy:

$$\mathcal{L}_{\mathrm{val}} = -\frac{1}{N} \sum_{n=1}^{N} \left( y_n^{\mathrm{val}} \log \hat{y}_n^{\mathrm{val}} + (1 - y_n^{\mathrm{val}}) \log(1 - \hat{y}_n^{\mathrm{val}}) \right) \tag{11}$$

Here, $N$ denotes the number of part latents, $y_n^{\mathrm{val}}$ is the ground-truth one-hot label indicating whether the $n$-th part is valid, and $\hat{y}_n^{\mathrm{val}}$ is the predicted validity score after sigmoid activation.

The total training objective is a weighted combination of all components:

$$\mathcal{L}_{\mathrm{total}} = \lambda_{\mathrm{skt}} \mathcal{L}_{\mathrm{skt}} + \lambda_{\mathrm{ext}} \mathcal{L}_{\mathrm{ext}} + \lambda_{\mathrm{rot}} \mathcal{L}_{\mathrm{rot}} + \lambda_{\mathrm{val}} \mathcal{L}_{\mathrm{val}} \tag{12}$$

In all experiments, we set the loss weights as $\lambda_{\mathrm{skt}} = 2$, $\lambda_{\mathrm{ext}} = 1$, $\lambda_{\mathrm{rot}} = 1$, and $\lambda_{\mathrm{val}} = 5$.

## A.3 Implementation Details for Baseline Methods

**DeepCAD:** For DeepCAD [42], we use its official open-source implementation. The model is first trained for 1000 epochs with the sequence reconstruction objective, jointly optimizing a sequence encoder and a latent decoder to reconstruct CAD sequences. In the second stage, the sequence encoder is frozen, and a 4-layer PointNet++ point cloud encoder [21] (as provided in the official implementation) is trained for 200 epochs using an MSE loss to align its output with the latent space. At inference time, input point clouds are encoded into latent vectors, which are then decoded by the pretrained decoder to generate CAD instruction sequences.

**DeepCAD\*:** Since the original DeepCAD adopts a contrastive learning approach, where the point cloud encoder is trained to align with sequence latents, it may suffer from information loss due to weak geometry-instruction supervision. We provide **DeepCAD\***, a variant trained to directly decode CAD instruction sequences from point cloud features. The decoder takes as input the extracted point cloud features and a learned constant embedding, and is supervised with cross-entropy loss over quantized instruction parameters. All remaining architecture (i.e., point cloud encoder and sequence decoder), optimization scheme, and testing protocol follow the original DeepCAD setup, with the model trained for 200 epochs.

**TransCAD:** For TransCAD [43], since the official implementation is not provided, we reimplement the framework based on the original paper details. The overall architecture consists of a point cloud encoder (4-layer PointNet++), a Loop-Extrusion Decoder (4-layer Transformer decoder with 3-layer MLP), two separate decoders (a 3-layer MLP for extrusion parameters and a 4-layer Transformer decoder for loop parameters), and a Loop Refiner for quantization offset regression. All hyperparameters follow the specifications described in the original paper. Following their training setup, we supervise CAD sequence types, quantized parameters, and continuous offsets for 200 epochs.

**HNC-CAD:** For HNC-CAD [13], we adopt the official open-source implementation and use their released pretrained codebook and model weights. Given the test CAD models, we convert them into the primitive-based representation adopted by HNC-CAD and formulate the task as a conditional generation problem. The predicted shapes are then evaluated using Chamfer Distance (CD) and Invalid Rate (IR) to assess geometric fidelity and modeling correctness.

**SECAD:** For SECAD [40], we use the official implementation and pretrained model in their open-source repository for testing. Given input point clouds, the model reconstructs CAD shapes, which are evaluated using CD for geometric fidelity and IR for modeling validity.

**Point2Cyl:** For Point2Cyl [15], we utilize the official open-source code and pretrained model to generate CAD models from input point clouds. As the method does not produce construction sequences, we evaluate its performance using CD and IR for fair comparison.

**ExtrudeNet:** For ExtrudeNet [14], since no pretrained model is provided, we train the model from scratch using the official codebase and follow their data preprocessing pipeline. The trained model is used to reconstruct CAD shapes from input point clouds, which are evaluated using CD and IR.

# B  Additional Methodological Insights

## B.1  Additional Details for Autoregressive Implicit Part Decomposition

In PartCAD, we perform implicit decomposition through an autoregressive scheme, enabling each part to correspond to a procedural modeling action while conditioning on previous steps to capture inter-part dependencies and avoiding the complexities of explicit geometric partitioning. Compared to one-shot alternatives that attempt to generate all part latents simultaneously, this design yields more stable outputs and avoids premature or invalid part proposals. Although autoregression inherently reduces parallelism and may introduce error accumulation, we observed no significant drifting in practice, aided by explicit supervision from the part discriminator and strong geometric priors from rotation-guided projections. This design thus provides a balanced trade-off between procedural fidelity and robustness, highlighting the value of autoregression as a principled mechanism for modeling sequential CAD construction processes.

## B.2  Additional Details for Adaptive Point Cloud Projection

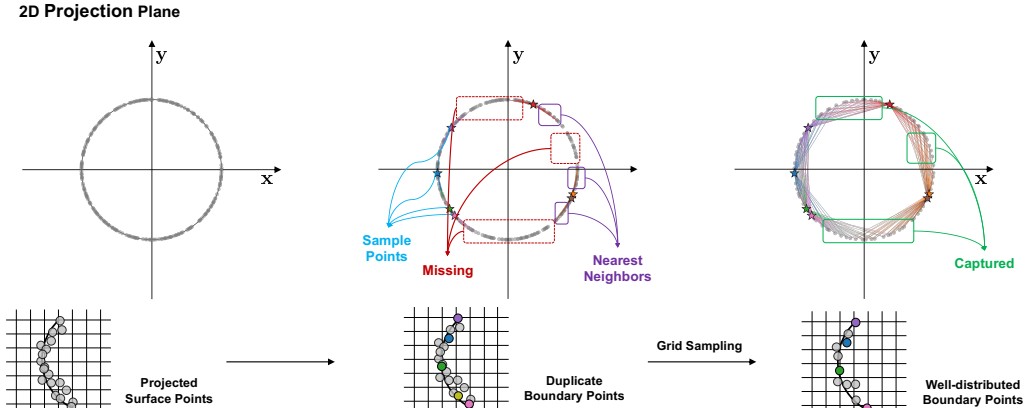

Figure 9: A toy example of an $xy$-plane projection of a cylinder CAD shape.

While normal filtering effectively suppresses points that are mostly irrelevant to the design intent in the current projection view, it remains insufficient to mitigate the clustering artifacts caused by projection distortion. In particular, points from side surfaces are often compressed toward projection boundaries, leading to over-concentration and reduced structural coverage. This degrades neighborhood quality in KNN-based aggregation and compromises the fidelity of geometric features.

To address this, we introduce an adaptive grid sampling strategy to spatially regularize the projection points. As illustrated in Figure 9, we visualize KNN neighborhoods for fixed sample points before and after grid sampling. Before processing, neighbors are overly concentrated within a local region, missing connections to global structural context. In contrast, grid-constrained resampling yields more spatially balanced neighborhoods, providing more geometric cues for downstream feature extraction. This strategy contributes to more reliable CAD instruction generation under projection guidance, as confirmed by ablation experiments.

## B.3  Additional Details for Hierarchical KNN Aggregation Kernel

While Euclidean-based KNN is widely used for point cloud feature extraction [22], it often struggles in projection spaces where points from geometrically distinct surfaces become spatially adjacent after flattening, especially in complex assemblies with nested or overlapping structures.

As illustrated in Figure 10 (left), we provide a toy example by projecting a hollow cylindrical CAD shape onto the $xy$-plane. In this scenario, naive Euclidean-based KNN (Figure 10, middle) produces spatially adjacent yet structurally inconsistent neighborhoods, resulting in limited receptive fields and unstructured pattern aggregation.

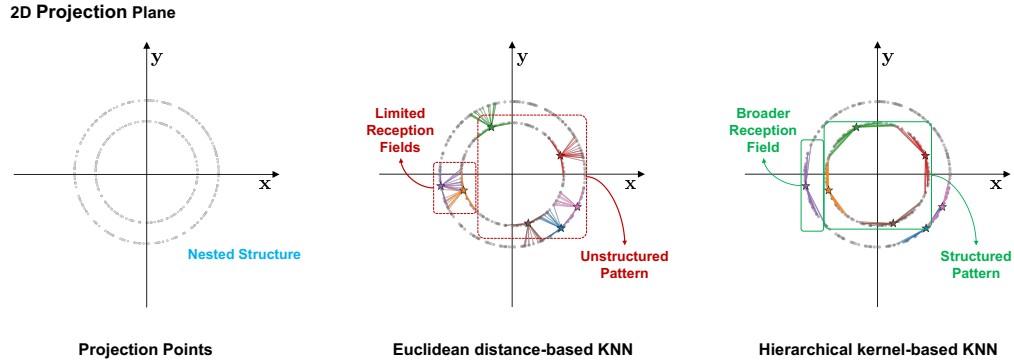

Figure 10: Euclidean distance-based KNN *vs.* hierarchical kernel-based KNN.

To address this, our hierarchical KNN aggregation kernel progressively refines neighborhood selection based on spatial distance, radial constraint, and normal similarity (see Equation 2). As shown in Figure 10 (right), compared to Euclidean-based KNN, our method forms more semantically consistent neighborhoods with broader receptive fields, better capturing the global CAD structure even in nested regions. This improvement strengthens projection-based feature learning and improves downstream CAD instruction generation.

## C    Additional Experiments

### C.1    Additional CAD Modeling Visualization

To further illustrate the part-aware generation process of PartCAD, we present step-by-step modeling visualizations in Figure 11. We incrementally decode the predicted part latent sequence, where each latent generates a sketch–extrusion pair that corresponds to a meaningful modeling operation. This process continues until an invalid latent is encountered, indicating the end of the sequence. As shown in the figure, PartCAD effectively extracts structural cues from the input point cloud and assembles a complete CAD model through a composition of part-wise instructions. It is worth noting that for part latents associated with sketch–extrusion pairs involving Boolean operations such as cut, join, and intersection, skipping the preceding solid construction steps can lead to modeling failures, as these operations rely on existing geometry to function properly.

In addition, we provide additional visualizations of reconstructed models from the generated CAD instructions to demonstrate the robustness of PartCAD in handling complex modeling scenarios, as shown in Figure 12 and Figure 13. These examples highlight two representative challenges in CAD reconstruction. Specifically, Figure 12 illustrates cases involving complex sketch structures, where the model successfully reconstructs detailed multi-contour and nested profiles. Figure 13 presents examples with complex topologies, including multi-body assemblies and solid modeling operations (e.g., cut, join, or intersection). These visual results further demonstrate the effectiveness of PartCAD in producing structurally coherent and geometrically faithful parametric CAD programs under diverse and challenging design conditions.

### C.2    Additional Ablation Study on Point Cloud Quality

**Impact of Surface Perturbations:** Reverse engineering from noisy or imperfect 3D scans is a common challenge in real-world applications. To evaluate the robustness of PartCAD under such conditions, we simulate surface-level perturbations by injecting controlled Gaussian noise into the input point clouds. This setup mimics typical data acquisition errors observed in practical scenarios, such as sensor noise, surface jitter, or minor misalignments.

Specifically, we simulate surface perturbations by adding Gaussian noise independently to each coordinate of the input point cloud. Let $\mathbf{p}_i \in \mathbb{R}^3$ denote a point in the raw point cloud, and its perturbed version $\tilde{\mathbf{p}}_i$ is given by:

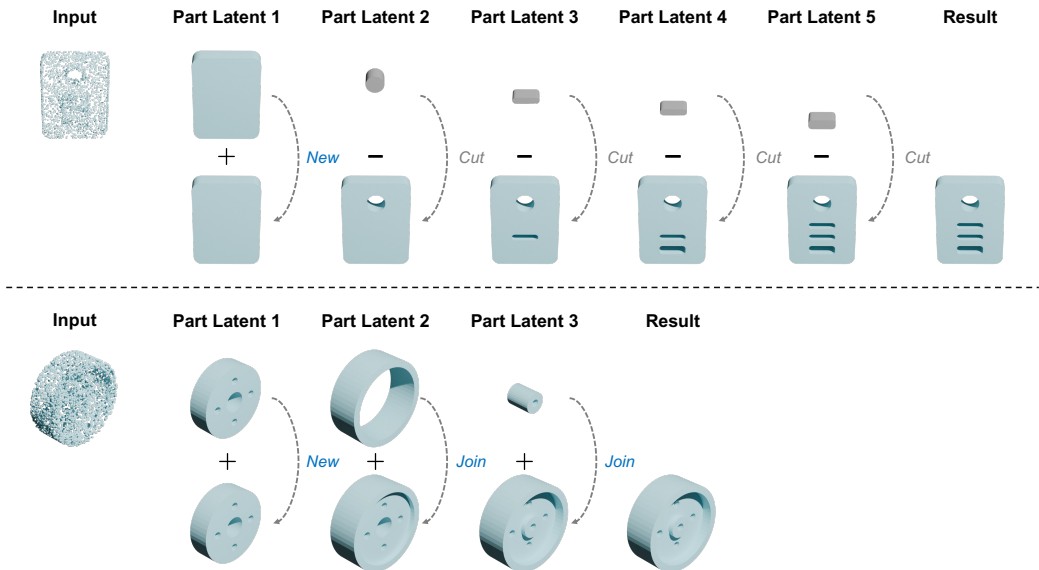

Figure 11: Visualization of part-wise latent decoding. Each column shows the cumulative result of decoding one additional latent, illustrating the progressive construction of the CAD model.

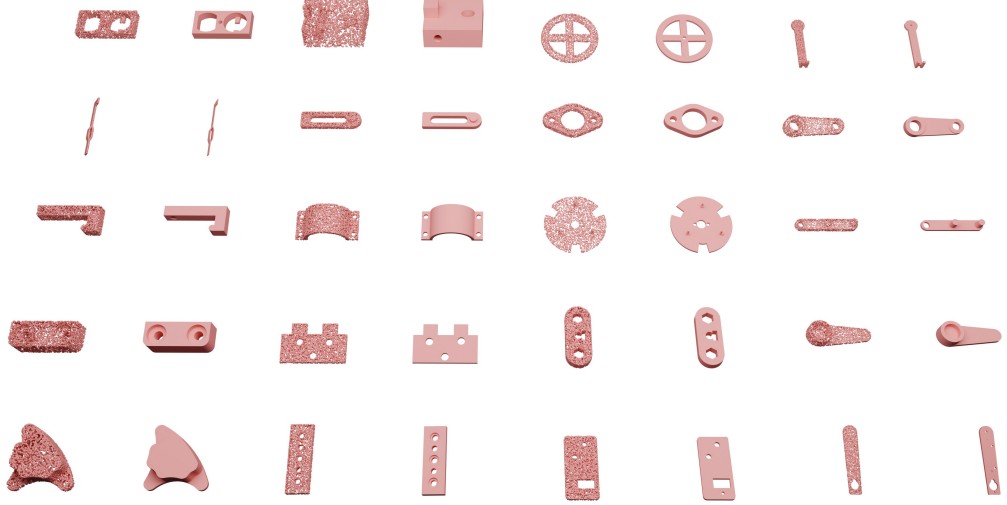

Figure 12: Qualitative visualization of reconstructed models with complex sketch structures. The examples highlight PartCAD's ability to recover intricate sketch layouts and maintain consistency across nested geometric constraints.

$$\tilde{\mathbf{p}}_i = \mathbf{p}_i + \boldsymbol{\epsilon}_i, \quad \boldsymbol{\epsilon}_i \sim \mathcal{N}(\mathbf{0}, \omega^2 \mathbf{I}). \tag{13}$$

Here, the noise $\boldsymbol{\epsilon}_i$ is sampled independently for each point and each dimension. The standard deviation $\omega$ is defined relative to the geometric scale of the point cloud as:

$$\omega = \xi \cdot \frac{\|\mathbf{b}_{\max} - \mathbf{b}_{\min}\|}{100} \tag{14}$$

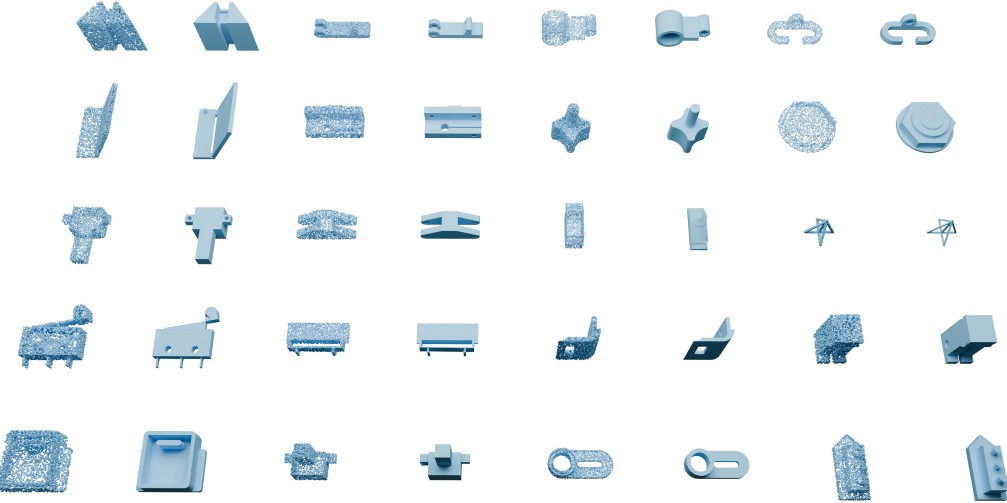

Figure 13: Qualitative visualization of reconstructed models with complex topologies. These examples showcase PartCAD's ability to handle multi-body relationships and solid modeling operations while maintaining structural coherence.

where $\|\mathbf{b}_{max} - \mathbf{b}_{min}\|$ denotes the diagonal length of the axis-aligned bounding box (AABB) of the point cloud with $\xi \in [0.5, 1.0, 2.5, 5.0]$ controlling the relative noise intensity.

Figure 14 presents qualitative visualizations of reconstructed CAD models generated from perturbed point clouds. We observe that PartCAD maintains strong performance under low to moderate perturbation levels (e.g., $\xi = 0.5, 1.0, 2.5$), successfully generating accurate and structurally consistent modeling instructions. Even under more severe noise conditions (e.g., $\xi = 5.0$), the framework demonstrates notable resilience, producing plausible results that largely preserve the semantic and geometric integrity of the original shapes. These results suggest that PartCAD is robust to realistic surface-level imperfections that commonly arise in practical 3D data acquisition settings.

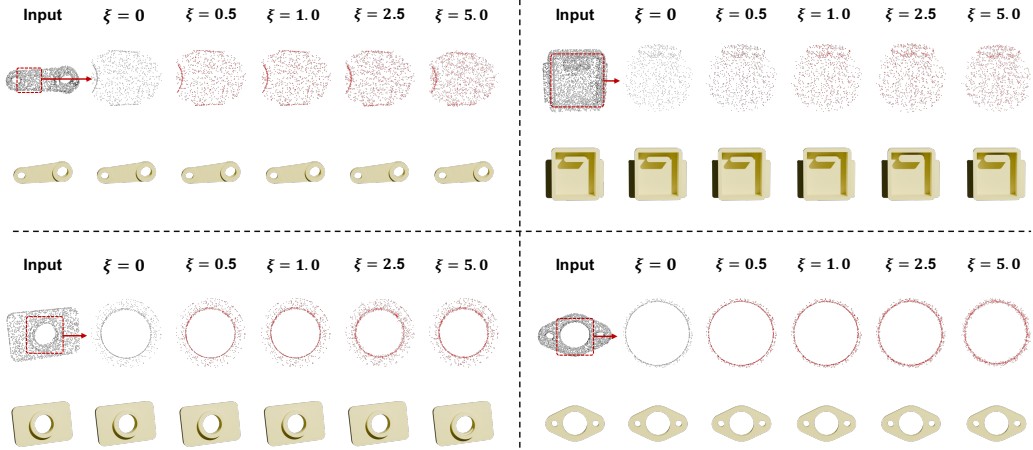

Figure 14: Qualitative visualization of reconstructed CAD models under different levels of Gaussian perturbation. We present four test examples from the DeepCAD dataset [42]. Each group of six columns corresponds to one test case. The first column shows the original input, with the raw input point cloud on top and the ground-truth CAD shape below. The remaining columns show results under noise levels $\xi = 0.5, 1.0, 2.5$, and $5.0$, where the perturbation is applied independently to each coordinate. The specific noise level used in each result is annotated above the corresponding column.

**Impact of Partial Inputs:** To assess the robustness of PartCAD under incomplete geometric input, we conduct an ablation study by randomly removing a portion of points from the input point clouds, simulating partial observations such as missing regions or occlusions. Specifically, we evaluate on the test set of the DeepCAD dataset, where 5%, 10%, 25%, and 50% of points are randomly removed and the resulting degraded point clouds are used for parametric instruction generation.

Figure 15 presents qualitative visualizations of the reconstructed models under different levels of incompleteness. We observe that PartCAD maintains strong modeling performance under mild to moderate point removal, consistently generating well-structured and semantically coherent CAD models. In some more challenging cases, however, substantial point loss (e.g., removal of 50% of input points) may obscure critical geometric cues, potentially resulting in degraded model quality or partial failure in instruction reconstruction.

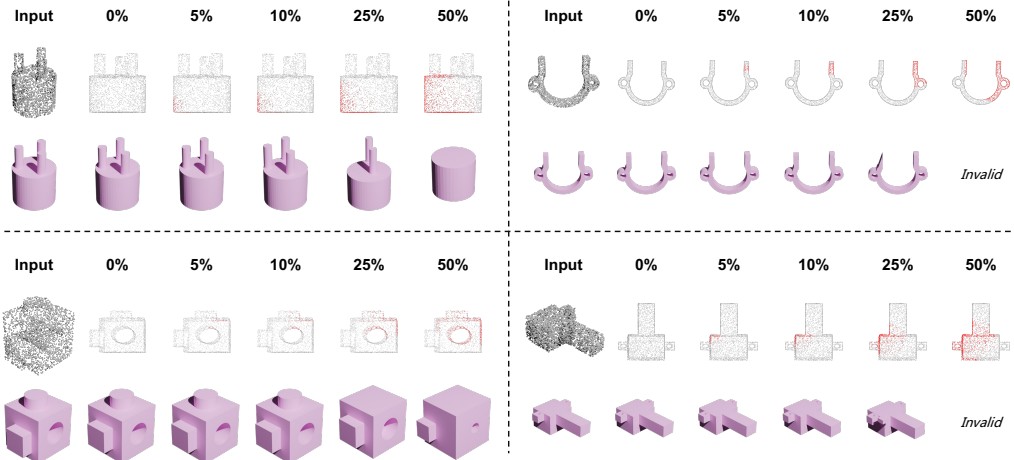

Figure 15: Qualitative visualization of reconstructed CAD models under varying degrees of point removal. We present four test examples from the DeepCAD dataset [42]. The first column in each block shows the original input, with the raw input point cloud on top and the ground-truth CAD shape below. The remaining columns show results under 5%, 10%, 25%, and 50% point removal (red points are removed). The percentage of missing points is indicated above each result.

## C.3 Additional Ablation Study Visualization

We present qualitative comparisons of different ablated variants in Figure 16, corresponding to the ablation study reported in Section 4. Compared to the full model, the ablated variants may produce geometric distortions, incomplete structures, redundant components, or imprecise modeling details (e.g., inaccurate extrusion distances). These issues highlight the importance of each module in ensuring accurate instruction generation and, therefore, faithful CAD modeling.

## C.4 Additional Statistical Robustness Experiments

To validate the robustness of our method, we further assess its performance consistency under repeated inference. Since model parameters are fixed after training, the main source of variability lies in inference randomness and stochastic point resampling in the Chamfer Distance computation. To quantify this effect, we performed 10 independent inference runs without fixed random seeds and report the mean and standard deviation of the main metrics on DeepCAD and Fusion 360 Gallery datasets. The results in Tables 6 and 7 confirm that our method remains stable and robust under varying inference iterations.

Table 6: Experimental results on the Fusion 360 Gallery [12], measured in terms of median CD and IR.

|      | Median CD ($\times 10^3$) | IR    |
|------|---------------------------|-------|
| Mean | 1.144                     | 1.321 |
| Std  | 0.041                     | 0.194 |

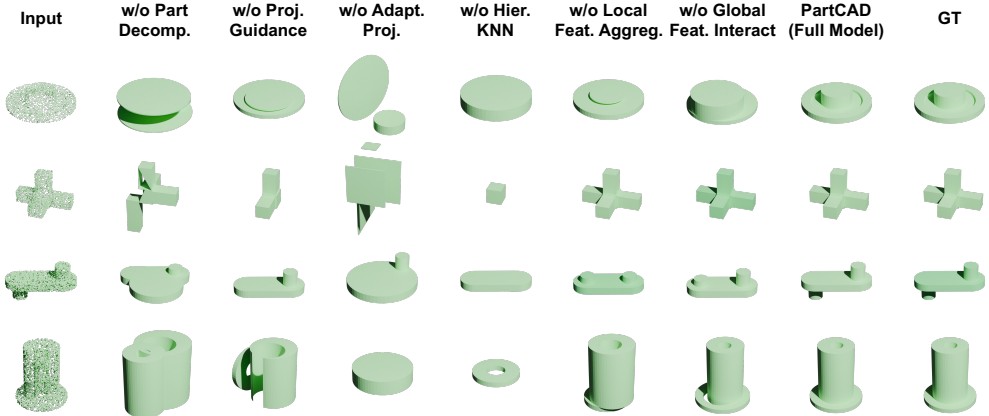

Figure 16: Visualization of ablation results. From left to right: input point cloud, reconstructed models by different ablated variants, and ground-truth CAD model. Each column shows the impact of individually disabling a core component.

Table 7: Statistical results on the DeepCAD dataset [42]. We report F1 scores for sketch parameters (Line, Arc, Circle) and extrusion parameters (Extrusion), CD (mean and median) for geometric fidelity, and IR for modeling validity.

| | F1 | | | | CD ($\times 10^3$) | | IR |
|---|---|---|---|---|---|---|---|
| | Line | Arc | Circle | Extrusion | Mean | Median | |
| Mean | 82.852 | 56.124 | 81.156 | 95.599 | 4.908 | 0.2373 | 0.911 |
| Std | 0.129 | 0.304 | 0.188 | 0.109 | 0.065 | 0.0009 | 0.038 |

# D   Failure Cases and Discussion

While PartCAD demonstrates strong performance in generating structured CAD models from point clouds, certain failure cases highlight the remaining challenges. We provide some representative failure cases in Figure 17, where PartCAD successfully reconstructs the overall shape and topology but exhibits some local errors, such as geometric displacement (left), a missing part (middle), and an incomplete curved region (right). Based on our experiments, these issues arise from two primary factors. First, converting continuous design parameters into discrete numerical representations can lead to modeling precision loss, particularly in fine-grained or tightly constrained structures. Second, complex or highly detailed sketches, especially those involving nested or intricate contours, can challenge the model's ability to capture localized geometric features.

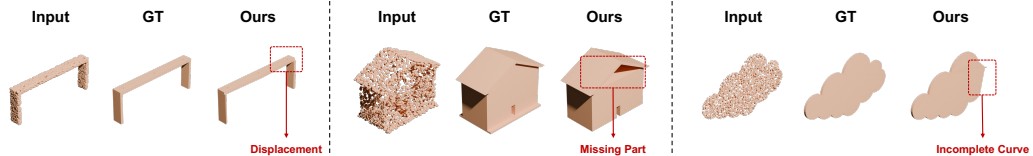

Figure 17: Visualization of three failure cases. While the global structure is preserved, minor local errors such as displacement, missing part, and incomplete profile can be observed.

Beyond the main contributions of this work, we also identify several insights that merit further discussion.

- **Modeling Ambiguity.** In real-world CAD design, a unique shape can be constructed through multiple valid modeling trajectories, reflecting diverse design philosophies and user preferences. Defining an *optimal path* typically requires additional criteria (e.g., minimizing sequence length, the number of sketch–extrusion pairs, or the use of specific primitives), which may diverge from natural human modeling habits. While our current framework adopts a deterministic decoding

strategy, generating distributional outputs that capture multiple plausible construction sequences remains a promising direction for future work. In addition, reward-based reinforcement learning paradigms may offer viable paths for addressing this challenge.

- **Intermediate representation.** Decomposing a complex shape into simpler intermediate representations is a widely accepted consensus in geometric learning, as demonstrated in voxel- and point-related tasks [57, 58]. Beyond the implicit decomposition in PartCAD, we also investigated more explicit schemes, such as bypass tasks that segment input points into semantically meaningful subsets or clustering parts based on feature similarity to provide stronger geometric grounding, but these efforts yielded only marginal (or no) accuracy gains while incurring additional computational and architectural complexity. However, we still believe that designing effective intermediate representations remains a promising direction, as a well-chosen abstraction could substantially simplify the generation of complex CAD modeling instructions.

- **Dataset Limitation.** The size, diversity, and generalization capacity of high-quality datasets remain a key bottleneck for building more powerful CAD generation models, especially when targeting complex modeling scenarios. For example, incorporating additional basic primitives (e.g., free-form curves) together with richer modeling operations (e.g., revolve, loft, chamfer) would allow trained models to better adapt to real-world design tasks, and several recent works have begun moving in this direction [59]. Moreover, current datasets exhibit long-tail distributions in model part counts and may suffer from out-of-distribution issues (e.g., discrepancies in shape scale or complexity from real-scan) both posing significant challenges to robust generalization in practical applications. Our framework is naturally extensible to enriched datasets, and future work will aim to exploit larger-scale corpora to further strengthen robustness and real-world applicability.

