# OpenReview forum: "Learning CAD Modeling Sequences via Projection and Part Awareness"
_NeurIPS.cc/2025/Conference — NeurIPS 2025 poster_

### Official Review · Reviewer_yQun · 2025-07-01

**Clarity:** 2
**Significance:** 2
**Originality:** 3
**Rating:** 5
**Confidence:** 4

**Summary:**

This paper introduces PartCAD, a semi-autoregressive framework designed to reconstruct CAD modeling sequences from 3D point clouds. The approach consists of (1) decomposing point clouds into part-aware latent representations; (2) a projection guidance module leveraging triplane projections to encode design intent; and (3) a non-autoregressive decoder that synthesizes sketch-extrusion parameters efficiently.

**Questions:**

1- Can you generate the CAD from sketch or image insdeat of point-cloud?

**Ethical Concerns:**

["NO or VERY MINOR ethics concerns only"]

**Final Justification:**

The authors have addressed my concerns, and the paper is very interesting and makes a significant contribution to AI and CAD. I am happy to increase my score.

**Limitations:**

1- Real-world applicability may be limited if input point clouds are very noisy or incomplete beyond tested scenarios.

2- The current implementation does not address ambiguity in the mapping from geometry to modeling sequence, nor does it quantify uncertainty or propose mechanisms to generate alternative plausible sequences.

**Quality:**

3

**Strengths And Weaknesses:**

# Weaknesses
1- Limited Scope of CAD Operations: The approach is primarily demonstrated for sketch-extrusion-based CAD modeling. More complex operations (e.g., sweeps, lofts, free-form surfaces) or richer command vocabularies are not addressed, limiting generality.

2- One-to-Many Mapping: The authors acknowledge the inherent ambiguity in mapping geometry to command sequences, but the paper offers little in terms of handling multi-modality or uncertainty (e.g., via distributional outputs or probabilistic modeling).

3- Limited exploration of alternative representation: The work is positioned between CSG, B-Rep, and sequence models but does not provide much insight on where it best fits, or its limits compared to modern neural B-Rep/CSG generative models.

# Strengths
1- Technical Contributions: The part-wise decomposition and projection-guided features are well-motivated. The semi-autoregressive architecture, especially the use of triplane projections and a hierarchical KNN kernel for better local feature aggregation, appears novel within this context.

2- Ablation Studies: The ablations highlight the necessity of the major architectural components, strengthening the empirical case for the proposed design.

---

> ### Author Rebuttal · Authors · 2025-07-30
>
> Thank you for your careful assessment and recognition of our novel contributions, such as the semi-autoregressive architecture, part-wise decomposition, triplane projection guidance, and hierarchical KNN aggregation. We appreciate your acknowledgment of our technical soundness and clear ablation studies. Please find below our responses to your concerns, organized by weaknesses, questions, and limitations.
>
> ---
>
> **W.1: Limited Scope of CAD Operations**
>
> While our experiments focus on sketch-extrusion operations (that constitute the majority of parametric CAD workflows), our framework is inherently operation-agnostic. The core contribution lies in projection-guided, part-aware geometry reasoning, which readily extends to other operations such as sweeping and lofting. Our approach introduces a learnable mapping from geometric features to modeling instructions that is not constrained to specific operation types. Extending to sweeps or lofts would require expanding the instruction vocabulary and incorporating additional primitives (guided curves, profiles), but the fundamental projection-guided reasoning and part-aware decomposition principles remain directly applicable.
>
> ---
>
> **W.2 / L.2: One-to-Many Mapping and Ambiguity Resolution**
>
> You raised an insightful point about the inherent ambiguity in geometry-to-sequence mapping. As discussed in the Limitation section of the main paper, multiple valid construction paths often exist for the same geometry, reflecting different design philosophies and modeling preferences.
>
> Our current deterministic decoding validates the proposed framework, but we recognize this limitation. In preliminary investigations, we found that resolving ambiguity requires defining optimization criteria (e.g., minimizing sequence length, the number of sketch-extrusion pairs, or the use of sketch primitives). However, these may not align with real-world practices, where experienced designers often prefer multiple simple operations over single complex ones for clarity and reusability.
>
> We strongly agree with your suggestion on distributional outputs. Generating multiple plausible instruction paths represents a promising direction that could provide designers with meaningful alternatives while preserving semantic richness. This could involve variational architectures, mixture models, or beam search strategies to explore valid reconstruction sequences.
>
> ---
>
> **W.3: Alternative Representation Exploration**
>
> Our work is strategically positioned within sequence modeling to reconstruct procedural CAD instructions from point clouds. Unlike neural B-Rep approaches (explicit surfaces/topology) or CSG methods (Boolean primitives), our sequential representation captures the temporal and logical structure of the modeling workflow itself.
>
> This distinction is crucial, as we preserve not just geometric outcomes but procedural knowledge of how geometry was constructed. Sequential representations naturally support part-level modifications, constraint editing, and design iteration workflows fundamental to parametric CAD practice.
>
> Our experiments also demonstrate competitive accuracy against recent fitting-based methods, further validating the practical effectiveness of our framework. Finally, the instruction representation can be straightforwardly compiled into B-Rep or CSG forms for compatibility with existing CAD engines, whereas the reverse conversion remains under-constrained and not directly attainable.
>
> ---
>
> **Q.1: Alternative Input Modalities**
>
> While our current work focuses on point cloud inputs for implicit part decomposition and projection-guided decoding, the framework is flexible. Core ideas of part-wise decomposition and projection-guided reasoning can extend to sketch or image inputs through modality-specific encoder adaptations. In particular, the projection-based decoding design is well-suited for integration with 2D sketch or image features. Such extensions would significantly broaden practical applicability for reverse engineering and design digitization.
>
> ---
>
> **L.1: Real-World Point Cloud Challenges**
>
> We acknowledge that noisy or incomplete point clouds are common in real-world applications. In our main paper (Section 4) and Appendix C.3, we evaluate the robustness of our method through real-scan experiments (using data from consumer-level 3D scanner), along with systematic tests involving surface perturbations (to simulate noise) and partial inputs (to simulate occlusion or incompleteness). The results show that our framework remains effective under such challenging conditions.

---

> > ### Comment · Reviewer_yQun · 2025-08-04
> >
> > The authors have addressed my concerns, and the paper is very interesting and makes a significant contribution to AI and CAD. I am happy to increase my score.

---

> > > ### Author Response · Authors · 2025-08-08
> > >
> > > Thank you for your thoughtful feedback and encouraging words. We’re pleased that our responses have addressed your concerns, and we sincerely appreciate your support and positive assessment.

---

### Official Review · Reviewer_817B · 2025-07-03

**Clarity:** 3
**Significance:** 3
**Originality:** 3
**Rating:** 5
**Confidence:** 2

**Summary:**

The paper introduces PartCAD, a semi-autoregressive framework that takes point cloud as input and generates structured CAD modeling instructions. Given a global point cloud feature, it first generates part latents in auto-regressive manner. The part latents are then used to form projected point features to guide the generation of CAD instruction parameters.

**Questions:**

See strengths and weaknesses section.

**Ethical Concerns:**

["NO or VERY MINOR ethics concerns only"]

**Final Justification:**

I have read the authors' rebuttal and other reviewers' comments. My stance remains the same regarding the novelty and that I do not see any major weakness in the paper.

**Limitations:**

Yes.

**Paper Formatting Concerns:**

No formatting concern.

**Quality:**

3

**Strengths And Weaknesses:**

I am not too familiar with the task, dataset, and related works. Reading the paper, I appreciate the following strengths:
1. The paper is relatively clear. The figures help to understand the proposed method. I also appreciate the output representation is explained in the Appendix.
2. The effectiveness of each components in the proposed method: 1). Part awareness decoding, 2). Projection guidance, 3). Instruction generation are systematically ablated.
3. Results show superior results. The authors also include results from real-world 3D printed CAD models.

I do not see any major weakness from the paper. There are some parts that I am not too clear:
1. Referring to Figure 3, in the part latent decomposition, the part latent is decoded into 3D point clouds which are then projected into three canonical planes according to the orientation of each part. How is the decoding process and how is the 3D point cloud supervised? Also, how is the rotation for each part supervised? Does the dataset provide such information?
2. From the results, it seems the CAD models are rather simplistic, e.g., mostly are brackets with only a few sketches and extruded components. How about CAD models with complex shapes and a lot of holes? Is it limited by DeepCAD dataset or there are underlying challenges to reconstruct complex CAD models? e.g., long generation sequence resulting in error accumulation or model capacity. What is the highest number of part latents the proposed method can generate?

---

> ### Author Rebuttal · Authors · 2025-07-30
>
> We sincerely appreciate the time and effort you devoted to reviewing our work. We are glad that you found the overall structure clear and that the figures and appendix supported the presentation effectively. Thank you for your positive comments on the design and evaluation of key components, including the part-aware decoding, projection guidance, and instruction generation, as well as your recognition of the real-world reconstruction results. Below please find the responses to the two points you raised for clarification.
>
> ---
>
> **W.1: Part Latent Decomposition Process and Supervision**
>
> We respond to this question in three aspects:
>
> (1) Part Latent Decomposition and Decoding: As described in the Autoregressive Implicit Part Decomposition part of Section 3, the input 3D point cloud is encoded and autoregressively decoded into several part latents via a part decoder (implemented as a four-layer Transformer). These part latents serve three purposes: (i) guiding final instruction generation, (ii) providing rotation cues for projection, and (iii) being supervised for part validity.
>
> (2) Orientation Prediction and Supervision: Each part latent is fed into the Orientation Predictor (implemented as a single-layer MLP) to estimate its rotation parameters, which guide the subsequent projection onto canonical planes. The predicted rotations are supervised using a cross-entropy loss ($L_\text{rot}$) computed from the ground-truth orientation parameters (Euler angles).
>
> (3) Validity Supervision: To supervise the implicit decomposition of the 3D point cloud, a Part Discriminator (implemented as a one-layer MLP) predicts the probability that each part latent corresponds to a valid part. A binary cross-entropy loss $L_\text{val}$ is computed between these predictions and the ground-truth validity labels.
>
> The supervision details for both rotation and validity are described in the Optimization part of the Section 3 and all supervision signals (including rotation parameters and part validity one-hot labels) can be directly obtained from the dataset.
>
> ---
>
> **W.2: Model Complexity and Scalability**
>
> Beyond the qualitative results presented in Figures 5–6 of the main paper, we also provide further examples in Figure 1 and Figures 12–13 (Appendix C.1), showcasing reconstruction results for CAD models with complex sketch structures and topologies. These examples highlight PartCAD’s capability to recover intricate sketch layouts and handle multi-body relationships and solid modeling operations. While longer instruction sequences may introduce greater prediction uncertainty, we have not observed significant error accumulation in practice. Empirically, modeling difficulty is influenced more by geometric complexity than by the number of part latents. For example, a geometrically complex single-part model can pose greater challenges than a multi-part assembly of simpler components.
>
> As detailed in the Implementation Details part of Appendix A.1, our current experimental setup supports up to 10 part latents (i.e., sketch–extrusion pairs) per CAD model. Each part latent can include up to 6 sketch loops, with each loop consisting of at most 15 curve primitives. This configuration covers all modeling cases in the DeepCAD dataset.

---

> > ### Comment · Reviewer_817B · 2025-08-04
> >
> > I have read the authors' rebuttal and other reviewers' comments. My stance remains the same regarding the novelty and that I do not see any major weakness in the paper. I have nothing to add on.

---

> > > ### Author Response · Authors · 2025-08-08
> > >
> > > Thank you for taking the time to review our paper and consider our rebuttal. We sincerely appreciate your comments, engagement and feedback.

---

### Official Review · Reviewer_6YMg · 2025-07-03

**Clarity:** 3
**Significance:** 3
**Originality:** 2
**Rating:** 4
**Confidence:** 4

**Summary:**

This paper proposes an autogressive part decomposition based pipeline for learning CAD Modeling Sequences. Central to the pipeline is an autoregressive encoder-decoder module that decomposes an input point cloud in to a sequence of part-level latent features. These features are combined (with some attention mechanism) with triplane projection guidance refined from projected point clouds, are then decoded into per part parameters which form the final CAD program. Qualitative examples and quantative numbers demonstrate that the proposed method clearly outperforms a couple well-known baselines for this problem.

**Questions:**

My main concerns for the paper centers around how generalizable and extendable this pipeline is. Granted, the key design is proven to work for the particular dataset and test cases, but I am very unsure whether this will work in more general cases, for the following reasons:

1. also branded as "autoregressive", the part decomposition pipeline lacks an explicit intermediate geometry representation, and instead just conditions on the previously generated latents. This, while fine for a bunch of coarse objects, can lead to problems when there exist fine grained geometries that require a sequence of precise operations. Moreover, as the sequences grow longer, it will be increasingly more difficult to reason about the part latents (especially when there are operations other than union) and the entire shape (and figure out what is left). Granted, the discriminator alleviate this a bit, but I still feel that a more principled solution is needed.
2. A related issue is that without intermediate correction and geometry representation, there can be potentials for error accumulation / drifting as the sequences grow longer. I didn't notice this too much in the provided examples, but still, more concrete analysis might help more.
3. CAD construction sequences are inherently ambiguous, and muliple paths exist for each shape. It seems very challenging for the current pipeline to handle this --- even if it is theoretically possible, I do suspect that the current method relies very strongly on similar examples seen in the training data, and will have issue in exploring other possible sequences, more so due to the unclear capacity of the decoder to actually perform geometric reasoning. Granted, this might not be an issue on common shapes with enough high quality data, but this might cause problems for other complex shapes that lack enough data (e.g. the hexagonal ring in Figure 7)
4. Step 3 also relies heavily on the quality of step 1 decompositions. If CAD instructions acutally need to span across parts identified in step 1, I don't think the current method can handle this. In other words, I think the method is kinda equivalent to a single step autoregressive part-by-part CAD decoder, just with projection features being injected in the middle of the layers.


I am still positive on the paper because it is effective for the current set of test cases. But explaning / providing insights to the questions above would further strengthen my take on this paper.

**Ethical Concerns:**

["NO or VERY MINOR ethics concerns only"]

**Final Justification:**

I still have concerns, sure. But this is a good paper and should be accepted. (will keep the 4 but this is close to a 5)

**Limitations:**

Yes

**Quality:**

3

**Strengths And Weaknesses:**

Strengths:
- Well engineered pipeline that delivers results that clearly outperforms relevant works. Many design choices clearly shout considerable effort in exploring possible design choices e.g. the steps used to refine the projected triplane features (ablations show that without these refinements, such projections don't help) and the specific use of attention in fusing the part latents and the geometric features.
- While the main paradigm (autoregressive decoding) and the particular model designs are fairly standard, the paper shows that with the right kind of data, feature representations and scaling (8 A100), such a pipeline can achieve very reasonable performances on this particular tasks without further, more complex inductive biases.
- Comprehensive evaluations, ablations and discussions.

Weaknesses:
- Novelty mainly lies in the way the components are combined (particularly, how and where to inject projection guidance), but the techniques used in the 3 main steps are all fairly standard, and thus limiting the novelty of the work a bit.
- Seems to have a strong reliance on having high quality training data in large scale, unclear if this paradigm is generalizable to more long-tailed part of the distribution (see questions)
- Many cad shapes naturally have ambiguity / multiple ways in how they are constructed. The current pipeline does not appear to have the capacity of handling that, and in addition, might suffer from drifting and other common issues of an autoregressive model due to the lack of an explicit intermediate geomtric representation.

---

> ### Author Rebuttal · Authors · 2025-07-30
>
> We greatly appreciate your detailed and thoughtful evaluation. Your positive assessment of our pipeline design, projection features, attention mechanisms, and comprehensive ablation studies is highly valued. Below we address your insightful concerns with additional clarifications.
>
> ---
>
> **W.1: Technical Novelty and Integration**
>
> While individual modules do build upon established techniques,  our main contribution lies in their thoughtful integration to address CAD-specific challenges. Our novelty centers on: (1) autoregressive part decomposition that produces interpretable, structure-aware latents as semantic anchors; (2) projection guidance that injects design-intent-aware features through rotation-guided triplane projections; and (3) non-autoregressive parameter decoding that efficiently generates sketch-extrusion instructions while preserving consistency. This cohesive design has demonstrated effectiveness and contributes to more interpretable CAD generation.
>
> ---
>
> **W.2/W.3/Q.3: Model Generalization and Data Dependency**
>
> You raised excellent points about data dependency. We acknowledge that high-quality data plays an important role, as in most deep learning domains. However, our framework goes beyond template memorization through structured part-wise decomposition and projection-guided decoding, offering generalization beyond replicating training sequences. We performed careful deduplication to prevent data memorization and ensure that the model doesn't rely on near-identical examples. As shown in Figure 1 (main paper) and Figures 12-13 (Appendix C.2), our method successfully reconstructs diverse shapes with complex sketch structures and intricate topologies, demonstrating robustness across various design variations, though extremely complicated configurations remain challenging.
>
> ---
>
> **W.3/Q.3: Construction Ambiguity and Modeling Diversity**
>
> We acknowledge that construction path ambiguity is a fundamental challenge in procedural CAD modeling, as the same geometry can often be constructed through different valid instruction sequences. Addressing such ambiguity often requires specifying some optimal criteria (e.g., minimizing sequence length, the number of sketch–extrusion pairs, or the use of curve primitives), which may diverge from natural human modeling habits. Rather than enforcing a single canonical path, our work focuses on how to effectively generate valid instruction sequences for the given shape. While our current framework employs a deterministic decoding strategy, future work will aim to model this ambiguity explicitly and explore criteria for defining optimal or human-aligned construction sequences.
>
> ---
>
> **W.3/Q.1: Intermediate Representations and Structural Supervision**
>
> We appreciate your insightful comments regarding the intermediate representations. In our current design, we adopt an autoregressive implicit part decomposition strategy conditioned on both the 3D point cloud features and previously generated latents. This approach is inspired by the incremental nature of CAD modeling, where even operations like Cut or Intersect can be viewed as negative geometric increments. To guide the decomposition process, we employ a Part Discriminator that evaluates the validity of each generated latent and provides supervision during training. Beyond this core motivation, we would like to share two of our experiences.
>
> First, for explicit intermediate representations, two common forms are part-level point cloud segments or part-specific point cloud features. We experimented with bypass tasks (e.g., part-level segmentation) to incorporate more structured intermediate cues. However, these efforts yielded only marginal gains in accuracy and came at the expense of increased computation and framework complexity. As a result, we opted for a more flexible implicit decomposition strategy coupled with part validity supervision, which has proven effective across a wide range of CAD modeling scenarios.
>
> Second, regarding the autoregressive generation strategy, we also experimented with one-shot approaches that generate all part latents from the input point cloud in a single forward pass. However, we observed lower performance and frequent discontinuities in latent generation (e.g., premature invalid part latents).
>
> Despite these experiences, we recognize the importance of intermediate representations and will continue to explore more effective and efficient mechanisms to incorporate them into future iterations of our PartCAD framework.
>
> ---
>
> **W.3/Q.2: Autoregressive Generation and Error Accumulation**
>
> We agree that error accumulation or drifting is a common challenge in purely autoregressive frameworks. In our preliminary experiments, we observed this issue when extending the autoregressive formulation to the CAD instruction generation stage (i.e., directly generating the sequence tokens). As discussed in W.3/Q.1, introducing more explicit intermediate representations helps alleviate this issue by providing stronger geometric grounding. Beyond that, we also explored feature-level solutions, such as similarity-based mechanisms for distinguishing part latents, but they yielded limited gains.
>
> In our current setup, where autoregression is applied only to the part decomposition stage, we did not observe significant drifting effects in our experiments, as evidenced by both quantitative results and visual reconstructions. In addition, several design choices help mitigate this issue. The Part Discriminator provides explicit supervision to guide the generation of valid parts and suppress the propagation of invalid latents. The rotation-guided projection introduces strong geometric priors during instruction decoding, while the localized and structured nature of sketch loops offers further geometric constraints across parts.
>
> ---
>
> **Q.4: Cross-Part Operations and Decomposition Dependencies**
>
> We understand your concern regarding potential limitations of the decomposition and decoding strategy and would like to provide the following clarification.
>
> First, in typical CAD modeling workflows, any cross-part operations (e.g., Join, Cut, or Intersect) are generally performed after a base body has been created. This modeling paradigm is also strictly followed in all CAD sequences within the DeepCAD dataset. Our framework explicitly models each CAD part as a sequence of sketch–extrusion pairs, with each part latent independently decoded into its corresponding instruction sequence. Each sketch–extrusion pair represents a complete and self-contained solid modeling operation (e.g., Newbody Creation, Cut, Join, Intersect). Importantly, these operations are executed incrementally and can only act on geometry that has already been constructed up to that point. No instruction is allowed to depend on future parts or not-yet-created geometry. This paradigm adheres to standard CAD modeling conventions and is consistent with the behavior of commercial CAD systems.
>
> Second, while the final CAD instructions in our framework are generated in a single forward pass, the parameter decoder leverages both local 2D projections and global 3D context. The global context is encoded via autoregressively generated part latents, which inherently capture structural dependencies between each modeling step. This design enables geometry-aware and part-specific predictions by modeling both intra-part details and inter-part relationships.
>
> We acknowledge the existence of certain scenarios (like lofting operations requiring simultaneous access to multiple parts) that fall beyond our current scope, and we're interested in developing hybrid formulations for more complex inter-part constraints in future work.

---

> > ### Comment · Reviewer_6YMg · 2025-08-08
> >
> > Thanks for the detailed clarifications and sorry for the late response.
> >
> > W1: To clarify: my general criteria for evaluting a systems paper is 1) individual component novelty; 2) non-trivial combination/adaptation of components; 3) result quality. I agree that this paper surely ticks (2) and (3), but lack of (1) still limit the potential impact of the paper a bit, especially when it comes to the more general audience beyond CAD modeling. Thus why I listed this as a "weakness", but still think that this is a good (and novel) paper.
> >
> > W2: Thanks for the examples. I'll trust your words on that, but it would still be helpful to add some more analysis in the revision e.g. showing nearest neighbors in training set (it's challenging to find a good distance function, but still better than nothing); more OOD examples on real life scans or other datasets
> >
> > W3Q3: I think there are two separate concerns: 1) some sequence might be more "human-like" and more friendly to downstream applications, not being able to find that is a limitation, but one can live with such limitation; 2) for something that is not deterministic by nature, using a deterministic training scheme will lead to issues when data is not clean enough, or when the test case is sufficiently different from those in the data, more discussion on this will be great.
> >
> > W3Q1: Acknowledged. Would help to discuss this more clearly in the revision --- it always help to know "what does not work (at least for now"). I am not sure if the current setup is actually what people perceive as "autoregressive", might worth double checking.
> >
> > W3Q2: Acknowledge. Might help to still try to identify if and where such issues happen. If projection guidance is the key, showing a few failure without projection guidance can deliver a strong message.
> >
> > Q4: This is a reasonable explanation. I suppose this then falls into the question above concerning the quality in the decomposition stage then.
> >
> > I think I am definitely more positive side post rebuttal. Thanks again for the answers.

---

> > > ### Author Response · Authors · 2025-08-08
> > >
> > > Thanks for your time and thoughtful feedback. We greatly appreciate your positive comments and are glad that our clarifications helped address your concerns. In the revision, we will include dataset analysis and OOD examples (W.2), and provide more discussions on deterministic training, intermediate representations, and the autoregressive design (W.3/Q.1/Q.3), as well as a few failure cases for the effect of projection guidance (W.3/Q.2). We hope these updates will enhance the paper.

---

### Official Review · Reviewer_ZqHg · 2025-07-06

**Clarity:** 3
**Significance:** 2
**Originality:** 2
**Rating:** 4
**Confidence:** 4

**Summary:**

This paper introduces a semi-autoregressive framework that reconstructs instructions to compose CAD models given raw point clouds. The algorithm begins by embedding the inputs into part-specific latent codes. It incorporates the sketch information through a view-aligned triplane projections and decodes precise sketch-extrusion instructions non-autoregressively in a single forward pass. The results show good performance on DeepCAD and Fusion360 benchmarks.

**Questions:**

- Could the paper explicitly specify what $c_n$ is? What's the length of this vector? Is it dynamic/variable or always the same?
- How is the BRep-Part correspondence modeled ? This is different than point cloud-part correspondence (asked earlier in weaknesses). A primitives-vs-part debate can be beneficial also.
- What are the advantages disadvantages in using an autoregressive encoder with a direct decoder?
- Why do we have a KNN-like clustering and not learning those 'hierarchical' operations?

**Ethical Concerns:**

["NO or VERY MINOR ethics concerns only"]

**Final Justification:**

I thank the authors for addressing the concerns. I maintain my score of acceptance.

**Limitations:**

Limitations are properly addressed. Although, the broader societal impact is not fully discussed.

**Quality:**

3

**Strengths And Weaknesses:**

### Strengths

- I like that the paper sees sketch modeling as a guidance and brings tools from 3D generative modeling like triplanes.
- The non-autoregressive decoder is probably efficient and can bring some internal consistency.
-  The guidance fuses part-level semantics with local/global point features and can help getting good accuracy.
- The paper demonstrates good performance in standard datasets.

### Weaknesses

- The introduction argues for structure and hierarchy but most of the contribution in this regard is in part modeling. This seems to be a common theme in the literature but is definitely simplistic. 3D models are so much more than parts. They are compositions of simple-to-complex primitives, each of which can have or not have a semantic meaning. Parts are only some of those basis functions. I think those claims should be tuned down and adjusted to the actual contributions.

- The idea of using different latent codes for parts is not new at all. In particular, it has been explored and popularized for example by the 3D capsule networks. I would suggest that the paper cites and explicitly mentions these:
  * Zhao et al. "3D point capsule networks." CVPR 2019.
  * Dubrovina et al. "Composite shape modeling via latent space factorization." ICCV 2019.

- The paper explicitly focuses only on sketch-extrude type models and utilizes sketches that are composed only of lines, arcs and circles. Although these constitute most of the operations / primitives, it is very unlikely that a single CAD model exists that is composed only of those operations / primitives. Recent CAD modeling tools have much more coverage and less restrictions than what is proposed here. This includes CAD-Recode (not cited):
  * Rukhovich et al. "CAD-Recode: Reverse Engineering CAD Code from Point Clouds". arXiv:2412.14042.

- Speaking of parts in CAD models is interesting. We can segment the point cloud into parts but in a CAD model, primitives can belong to different parts causing non-uniqueness or inconsistencies. It seems like there is no explicit treatment of this in the current paper. Could the authors elaborate on how the model handles these cases?

- I see that the overall pipeline is similar to Point2Cyl, except that a rather new toolset is used. I think the paper should discuss similarities / differences. In particular, how the discrete state of tokens would compare to a continuous model as in Point2Cyl is certainly interesting.

- The method is counter-intuitive to me. This is not negative per se but requires clarification. I would expect that the point latents are generated by a one-step encoder whereas the decoder could be auto-regressive (like in LLMs). Why does the paper argue for the reverse? What are the advantages disadvantages? The paper gives some explanations but not a comparative one.

- Rotations are regressed over the discrete tokens. Wouldn't this cause problems / errors? I feel like all canonicalizations would be a bit off.

- What about the CC3D-ops dataset which contains longer (operation) sequences? I think we have more datasets to evaluate on compared to DeepCAD and Fusion360 now.

- The evaluation metrics are fine but at this stage I would really recommend adopting a CAD-aware loss that is evaluated on instructions / operations and their parameters. This would be a true 'program-distance', that characterizes how effort is needed to turn a sequence of operations into another one.

- In Table 1, I think Point2Cyl can be included as their sketches are more general and can be easily parsed into arcs/lines etc.

- Real scan experiments are limited to qualitative ones. Can we have some quantitative evaluations for known CADs?

- The checklist claims that the error bars included, however non of the tables or plots provide error bars, and authors ask the reader to 'trust' their experience that the results are stable. This is unacceptable to me. My current rating is 4 but if the rebuttal does not address this, I will be lowering the score.

---

> ### Author Rebuttal · Authors · 2025-07-30
>
> We sincerely thank you for the thorough review and constructive feedback. We appreciate your thoughtful questions that will help improve our work. Below are our responses to your concerns, along with additional clarifications.
>
> ---
>
> **W.1/W.4/Q.2: Structure, Parts, and CAD Modeling**
>
> We appreciate your point about 3D structural complexity. Our work specifically targets the procedural structure inherent to CAD modeling workflows, where part-wise decomposition provides a tractable abstraction that aligns well with standard CAD practices. Each "part" in our context represents a meaningful modeling step (sketch-extrusion pair) that captures design intent rather than purely geometric segmentation.
>
> We perform implicit decomposition through autoregressive latent generation, allowing each part to naturally correspond to procedural modeling actions while avoiding the complexities of geometric partitioning. Through our experiments, we found this implicit formulation more effective and scalable than explicit segmentation approaches. We will revise the introduction to better articulate this CAD-specific scope and procedural focus.
>
> ---
>
> **W.2/W.5: Related Work and Comparisons**
>
> Thank you for highlighting these important references that we missed. We will include them in the revision, as they provide valuable context for part-based latent modeling. Our work focuses specifically on procedural CAD sequence generation rather than static shape analysis or segmentation. Regarding Point2Cyl, our approach takes a different direction. We generate structured instruction sequences using discrete tokens aligned with formal grammar, ensuring syntactic validity and semantic interpretability. Point2Cyl, in contrast, fits continuous cylinders using implicit functions, without the symbolic structure needed for procedural modeling. We will include a more detailed comparison highlighting these complementary approaches.
>
> ---
>
> **W.3: CAD Operation Coverage**
>
> You raised an important point about our current scope. We acknowledge that real CAD models typically involve more operations and primitives. Our current work focuses on sketch-extrusion operations that constitute the majority of parametric CAD workflows, but this is indeed a limitation we should discuss. We appreciate the pointer to CAD-Recode and will cite it along with a discussion on extending to broader CAD operations. Our framework is designed to be extensible to richer operations (sweeping, lofting) and sketch primitives (splines, complex curves) given appropriate training data, which we plan to pursue in future.
>
> ---
>
> **W.6/Q.3: Autoregressive Design Choice**
>
> While global features are extracted in a single pass using a standard encoder, part latents are generated autoregressively to reflect the sequential nature of CAD modeling, allowing each part to condition on previous ones and capture inter-part dependencies consistent with real-world workflows. We did explore one-shot alternatives that generate all latents simultaneously, but observed reduced performance and less stable outputs, particularly with premature or invalid part generation. While autoregression does reduce parallelism and could potentially introduce error accumulation, we found minimal runtime overhead in practice and no significant output drift thanks to the supervision from our Part Discriminator. We will include a more detailed discussion of this design trade-off in our revision.
>
> ---
>
> **W.7/W.8: Discretization and Datasets**
>
> Discretization does introduce quantization error, but we found this trade-off worthwhile as it significantly improves training stability compared to direct regression approaches. The discrete formulation also enables more structured supervision and token-based evaluation. Regarding datasets, we focused on current datasets to maintain consistency with prior work. They contain complex samples with long, hierarchical sequences, suitable for testing our method's robustness. We appreciate the suggestion about CC3D and are currently applying for access to include it in future evaluations.
>
> ---
>
> **W.9: CAD-Aware Evaluation**
>
> We agree that CAD-aware evaluation is crucial. Our framework does integrate CAD-aware design in both training and evaluation. During training, we supervise both geometric parameters and structural/semantic tokens that encode high-level CAD concepts like boundary markers and Boolean operations using cross-entropy loss. Our evaluation goes beyond pure geometry through token-level F1-scores that assess both low-level parameters and high-level instruction structure.
>
> ---
>
> **W.10/W.11: Point2Cyl Comparison and Real Scan Evaluation**
>
> We excluded Point2Cyl from instruction-level comparison because its outputs use continuous implicit representations rather than discrete symbolic primitives. While its 2D sketches may visually resemble our primitives, they're defined through fitted parameters rather than explicit modeling commands. Post-hoc parsing into symbolic tokens is possible but often unreliable. For real-world scans, our evaluation is primarily qualitative, as most real scan datasets unfortunately lack the high-quality parametric CAD annotations needed for rigorous quantitative comparison. We plan to curate more scan-CAD pairs with parametric labels for future research.
>
> ---
>
> **W.12: Statistical Rigor and Error Reporting**
>
> We sincerely apologize for the mismatch in our checklist, which was our oversight. You’re absolutely right that empirical results should be supported by proper statistics. Since model parameters are fixed after training, variability primarily arises from inference randomness and stochastic point sampling in the Chamfer Distance computation. To address this concern, we conducted 10 independent inference runs without fixed random seeds and computed mean and standard deviation for the main metrics (see the following 2 tables). Our original experiments followed strict protocols: consistent data splits, fixed training seed, dataset deduplication, and full implementation details (see Section 4, Appendix A.2–A.3). We acknowledge the importance of reporting statistical variation. We will include error bars and statistical analysis in the final version and sincerely appreciate your attention to this crucial issue.
>
> ---
>
> **Table: Experiments in DeepCAD Dataset**
>
> | Instance | F1 Line | F1 Arc | F1 Circle | F1 Extrusion | Mean CD | Median CD | IR   |
> |----------|---------|--------|-----------|--------------|----------------|-----------|------|
> | 1        | 82.97   | 56.66  | 80.79     | 95.62        | 4.87           | 0.238     | 0.89 |
> | 2        | 82.71   | 56.28  | 81.35     | 95.63        | 4.91           | 0.236     | 0.90 |
> | 3        | 82.64   | 55.76  | 81.11     | 95.68        | 4.89           | 0.238     | 0.86 |
> | 4        | 82.96   | 56.22  | 81.31     | 95.78        | 4.99           | 0.237     | 0.92 |
> | 5        | 82.93   | 55.76  | 81.14     | 95.59        | 4.84           | 0.238     | 0.95 |
> | 6        | 82.66   | 55.88  | 81.49     | 95.54        | 4.89           | 0.237     | 0.97 |
> | 7        | 82.85   | 56.48  | 81.04     | 95.44        | 4.80           | 0.237     | 0.93 |
> | 8        | 83.01   | 56.37  | 81.15     | 95.69        | 4.97           | 0.237     | 0.95 |
> | 9        | 82.94   | 55.96  | 80.99     | 95.40        | 4.90           | 0.236     | 0.85 |
> | 10       | 82.85   | 55.87  | 81.19     | 95.62        | 5.02           | 0.239     | 0.89 |
> | **Mean** | 82.852  | 56.124 | 81.156    | 95.599       | 4.908          | 0.2373    | 0.911 |
> | **Std**  | 0.1290  | 0.3040 | 0.1875    | 0.1086       | 0.0645         | 0.0009    | 0.0378 |
>
> ---
>
> **Table: Experiments in Fusion360 Dataset**
>
> | Instance | Median CD | IR   |
> |----------|------------------|------|
> | 1        | 1.15             | 1.63 |
> | 2        | 1.08             | 1.09 |
> | 3        | 1.14             | 1.51 |
> | 4        | 1.16             | 1.48 |
> | 5        | 1.11             | 1.42 |
> | 6        | 1.22             | 1.20 |
> | 7        | 1.10             | 1.46 |
> | 8        | 1.13             | 1.08 |
> | 9        | 1.15             | 1.27 |
> | 10       | 1.20             | 1.07 |
> | **Mean** | 1.144            | 1.321 |
> | **Std**  | 0.0408           | 0.1942 |
>
> ---
>
> **Q.1: Instruction Format Details**
>
> CAD modeling instructions are structured as $C = ${$c_n $}$_{n=1}^N$, where $N$ is the sketch–extrusion pairs. Each instruction consists of sketch and extrusion components, both quantized as 8-bit discrete sequences (see Section 3 and Appendix A.1). The sketch component has variable length depending on the number of 2D primitives and structural boundary tokens, while the extrusion component is fixed at 11 tokens covering orientation, position, scale, depth, and Boolean operations. For parallelization and efficient training, each instruction sequence is padded to a fixed length of 110 tokens, which covers all samples in our current dataset.
>
> ---
>
> **Q.4: Hierarchical KNN Design Rationale**
>
> The hierarchical KNN aggregation serves as an intermediate step for selecting reliable neighborhoods among 2D projection points, which are then processed by learnable encoders (EdgeConv layers). This design augments learning with geometric priors rather than replacing it entirely. It is motivated by the observation that important CAD structures (sketch profiles, extrusion contours) exhibit locally coherent patterns in projection views. Our hierarchical KNN applies three-stage refinement considering spatial proximity, radial alignment, and surface normal similarity to identify geometrically consistent regions that correspond to underlying modeling processes. This helps the encoder learn more robust features for CAD instruction generation.

---

> > ### Comment · Reviewer_ZqHg · 2025-08-06
> >
> > I thank the authors for addressing the concerns. I maintain my score of acceptance.

---

> > > ### Author Response · Authors · 2025-08-08
> > >
> > > We sincerely thank you for your detailed review and insightful comments. Your feedback helped us refine the scope and clarify several key aspects of our work. We appreciate your recognition and are glad to have your support for acceptance.

---

### Note · Authors · 2025-08-15

Dear all reviewers and AC,

We sincerely thank you for your time, constructive feedback, and valuable suggestions. Your insightful comments have helped us clarify our technical contributions, strengthen the experimental analysis, and improve the overall presentation.

In this paper, we present PartCAD, a semi-autoregressive framework for reconstructing structured CAD modeling instructions from 3D point clouds through projection-guided, part-aware geometry reasoning. PartCAD autoregressively decomposes raw geometry into part-aware latents enriched with triplane projection cues, and decodes them in a single pass into precise sketch–extrusion parameters. With the proposed adaptive projection refinement, hierarchical KNN feature aggregation, and projection-guided decoding, PartCAD delivers superior accuracy, robustness, and cross-domain generalization, offering new perspectives for interpretable CAD reconstruction and automation.

During the rebuttal process, we carefully addressed concerns regarding dataset scope, reconstruction ambiguity, evaluation methodology, and the coverage of CAD operations. In addition to clarifying these points, we engaged in discussions on the CAD modeling task and the underlying methodological choices, which not only strengthened the current submission but also inspired promising future research.

We greatly appreciate the constructive dialogue throughout the review process and the opportunity to refine and improve our work. We hope that our responses and clarifications have satisfactorily addressed the key concerns, and we look forward to further advancing this research direction. Thank you again for your thoughtful engagement with our submission.

---

### Decision · Program_Chairs · 2025-09-17

**Decision:**

Accept (poster)

**Comment:**

This paper introduces a part-aware framework that reconstructs CAD instructions from raw point clouds. The algorithm first embeds the inputs into part-specific latent codes. It decodes precise sketch-extrusion instructions non-autoregressively in a single forward pass. The results show good performance on DeepCAD and Fusion360 benchmarks.

All reviewer were positive before the rebuttal. The authors addressed minor issues raised by the reviewers during rebuttal. The scores remain positive after rebuttal. There is consensus that this paper should be accepted. The AC agrees with this decision.